



# Asymmetric impact of groundwater use on groundwater droughts

Doris E Wendt[1], Anne F Van Loon[1], John P Bloomfield[2], and David M Hannah[1]

[1]University of Birmingham, Birmingham, UK
[2]British Geological Survey, Wallingford, UK

**Correspondence:** Doris Wendt (dew637@bham.ac.uk)

**Abstract.** Groundwater use affects groundwater storage continuously, as the removal of water changes both short-term and long-term variation in groundwater level. This has implications for groundwater droughts, i.e. a below-normal groundwater level. The impact of groundwater use on groundwater droughts remains unknown. Hence, the aim of this study is to investigate the impact of groundwater use on groundwater droughts adopting a methodological framework that consists of two approaches.

The first approach compares groundwater monitoring sites that are potentially influenced by abstraction to uninfluenced sites. Observed groundwater droughts are compared in terms of drought occurrence, magnitude, and duration. The second approach consists of a groundwater trend test that investigates the impact of groundwater use on long-term groundwater level variation. This framework was applied to a case study of the UK. Four regional water management units in the UK were used, in which groundwater is monitored and abstractions are licensed. The potential influence of groundwater use was identified on the

basis of relatively poor correlations between accumulated standardised precipitation and standardised groundwater level time series over a 30-year period from 1984 to 2014. Results of the first approach show two main patterns in groundwater drought characteristics. The first pattern shows an increase of shorter drought events, mostly during heatwaves or prior to a long drought event for influenced sites compared to uninfluenced sites. This pattern is found in three water management units where the long-term water balance is generally positive and annual average groundwater abstractions are smaller than recharge. The second

pattern is found in one water management unit where temporarily groundwater abstractions exceeded recharge. In this case, groundwater droughts are lengthened and intensified in influenced sites. Results of the second approach show that nearly half of the groundwater time series have a significant trend, whilst trends in precipitation and potential evapotranspiration time series are negligible. Detected significant trends are both positive en negative, although positive trends dominate in most water management units. These positive trends, indicating rising groundwater levels, align with changes in water use regulation.

This suggests that groundwater abstractions have reduced during the period of investigation. Further research is required to assess the impact of this change in groundwater abstractions on drought characteristics. The overall impact of groundwater use is summarised in a conceptual typology that illustrates the asymmetric impact of groundwater use on groundwater drought occurrence, duration, and magnitude. The long-term balance between groundwater abstraction and recharge appears to be influencing this asymmetric impact, which highlights the relation between long-term and short-term sustainable groundwater

use.



## 1 Introduction

Groundwater is an essential source of water supply, as it provides almost half the global population with domestic water (Gun, 2012), 43% of the irrigation water (Siebert et al., 2010), and 27% of industrial water use (Döll et al., 2012), as well as sustaining ecologically important rivers and wetlands (de Graaf et al., 2019). The usage and dependency on groundwater resources has grown in the past decades (Famiglietti, 2014), particularly during meteorological droughts, when groundwater is used frequently (Taylor et al., 2013; AghaKouchak, 2015).

Meteorological droughts propagate through the hydrological cycle and the deficit in precipitation results in a groundwater drought (Wilhite, 2000; Van Lanen, 2006), defined as below-normal groundwater levels that are associated with short-term reductions in storage (Chang and Teoh, 1995; Tallaksen and Van Lanen, 2004; Mishra and Singh, 2010). Increased use of groundwater before or during droughts can also lower groundwater levels and can thereby aggravate groundwater droughts (Wada et al., 2013; Christian-Smith et al., 2015). Managing groundwater use during droughts is therefore important, as over-exploitation of groundwater has disastrous consequences (Custodio, 2002; Famiglietti, 2014; Russo and Lall, 2017; Mustafa et al., 2017). However, to date groundwater droughts have been studied under primarily near-natural conditions and there is limited conceptual understanding of the impact of groundwater use on groundwater droughts despite this being of interest to water regulators and policy makers.

Under near-natural conditions, the propagation of meteorological droughts to groundwater droughts depends on the antecedent condition of the land surface, subsurface controls on recharge, and non-linear response of groundwater systems (Peters et al., 2006; Tallaksen et al., 2009; Eltahir and Yeh, 1999). These processes determine the spatial distribution, duration, magnitude, and recovery of near-natural groundwater droughts (Van Lanen et al., 2013; Van Loon, 2015; Parry et al., 2018). However, in human-modified environments, groundwater droughts are also impacted or driven by water use (Van Loon et al., 2016b). This type of groundwater drought is therefore distinguished from a natural drought and referred to as human-modified or human-induced drought (Van Loon et al., 2016a).

In human-modified environments, understanding the influence of groundwater use on groundwater drought requires information related to both natural propagation of a drought and groundwater abstraction. Droughts can be influenced by historical and recent abstractions, as these change both short-term and long-term groundwater storage (Gleeson and Richter, 2017; Thomas and Famiglietti, 2015; Jackson et al., 2015). Unfortunately, information on groundwater abstraction, if available at all, is often considered commercially confidential. Abstraction records are usually unavailable for research, although these records are an important component of groundwater models that are developed for commercial and regulatory purposes (Shepley et al., 2012). Consequently, qualitative information about groundwater use and management regulations often is invaluable to investigate the influence of groundwater abstraction on groundwater droughts (Döll et al., 2014; Panda et al., 2007). Management regulations are organised on regional or even national scale. This scale differs from the much smaller scale at which groundwater droughts are often studied. For example, physically-based groundwater models, which are developed for regulatory purposes or research, rarely cover the entire drought-impacted area or the entire aquifer that is affected by a drought event (Peters et al., 2006; Tal-





laksen et al., 2009; Shepley et al., 2012). Studying groundwater droughts in human-modified environments would thus require
a regional approach to align the scale of a groundwater drought study with the scale at which management decisions are made.

The aim of this study is to investigate the impact of groundwater use on regional groundwater droughts in the absence of actual abstraction data. For doing so, a methodological framework is designed to investigate groundwater droughts in water management units under a broad range of conditions, i.e. from where groundwater use is a small proportion of the long-term annual average recharge to where it is (temporarily) a significant proportion of the long-term annual average recharge. A case
study from the United Kingdom (UK) is used, consisting of four water management units over the two main aquifers in the UK. As is common elsewhere, no data is freely available on actual abstractions for the four water management units in the case study area. However, information indicating the annual maximum abstraction according to the groundwater abstraction licence is available and groundwater level observations are provided for 170 sites in the four water management units. Consequently, inferential approaches are used to assess the potential impact of abstraction on groundwater droughts. Here, we used two
complementary approaches. Firstly, given the typically good correlation between precipitation and groundwater level time series under near-natural conditions (Bloomfield and Marchant, 2013; Bloomfield et al., 2015; Kumar et al., 2016), we used correlations defined by a limited number of near-natural groundwater hydrographs as reference. Deviations from this reference correlation are then used to qualitatively subdivide sites in uninfluenced and influenced by abstraction, and characterise the modifying effect of groundwater abstraction on regional groundwater droughts. Secondly, we investigated long-term effects of
abstraction through the spatial distribution of trends in groundwater level time series in relation to the distribution of licensed abstractions. The results are discussed in terms of the role groundwater abstraction plays in modifying near-natural groundwater droughts. A conceptual figure is proposed suggesting that long-term groundwater abstraction may modify drought frequency, duration, and magnitude depending on the proportion of abstraction and recharge.

## 2 Study area

The UK case study consists of four water management units across the Chalk and Permo-Triassic sandstone aquifers that are the two main aquifers in the UK (Figure 1). The two aquifers have contrasting hydrogeological characteristics. Regional groundwater flow and storage in the Chalk aquifer are dominated by its primary fracture network (Bloomfield, 1996) and secondary solution-enhanced fractures (Downing et al., 1993; Maurice et al., 2006). The response of Chalk groundwater hydrographs to driving meteorology is a function of regional variations in the nature of the fracture network, extent of karstification, nature of
overlying superficial deposits amongst other factors (Allen et al., 1997). In the Permo-Triassic sandstone aquifer, groundwater flow and storage are influenced by variations in the matrix porosity, variable aquifer thickness, and to a lesser extent by fracture characteristics (Shepley et al., 2008; Allen et al., 1997). Faults divide the Permo-Triassic sandstone in separate sections. The effect of faults varies widely. Some faults act as hydraulic barriers whilst others enhance permeability resulting in increased recharge (Allen et al., 1997). Hydrographs in the Permo-Triassic sandstones typically respond more slowly to driving meteo-
rology than those in the Chalk (Bloomfield and Marchant, 2013) and are influenced by local variation in aquifer thickness and confinement by superficial deposits.





**Table 1.** Regional features of the water management units summarising the 1) area size, 2) long-term precipitation (P) and potential evapotranspiration (PET) as calculated by Mansour and Hughes (2018), 3) hydrogeological features and 4) main groundwater use. All water management units are shown in Figure 1. In Figure S1, the purpose and locations of recent abstraction licences are shown. Hydrogeological information and groundwater use is based on Allen et al. (1997) and complemented with additional references (see last column).

| Water management unit & number of monitoring wells | Area (km$^2$) | Annual average (mm/yr) | Hydrogeological features | Groundwater use | Additional literature |
|---|---|---|---|---|---|
| 1: Lincolnshire | 1310 | P: 589 | Highly permeable outcrop due to dissolved fractures and weathering | Abstraction peaked in 1970 and reduced since 2000 | Whitehead and Lawrence (2006) |
| 38 wells | | PET: 454 | South-East of aquifer increasingly confined by superficial deposits | Abstractions exceed average recharge only during droughts | Bloomfield et al. (1995), Hutchinson et al. (2012) |
| 2: Chilterns | 1650 | P: 674 | Chalk aquifer partly covered by superficial deposits | Abstractions increased during 1970-2003 and decreased after 2003 | Jones (1980), Jackson et al. (2011) |
| 45 wells | | PET: 485 | karstification in valleys | recent abstraction is estimated on 50% of average recharge | Environment Agency (2010) |
| 3: Midlands | 1100 | P: 630 | Varying aquifer thickness from 120-300m | Abstraction exceeded the average recharge rates by 25% in 1980-90 | Zhang and Hiscock (2010) |
| 36 wells | | PET: 476 | Confined by superficial deposits in the East | Abstraction reduced in 2000 to meet average recharge | Shepley et al (2008) |
| 4: Shropshire | 1400 | P: 722 | Highly variable aquifer thickness: 30-1400m | Abstraction represented 40-50% of recharge in 1970-90 and reduced after 2000. | Cuthbert (2009), Voyce (2008) |
| 51 wells | | PET: 471 | Major faults interrupt groundwater flow across sandstone layers | River augmentation scheme increases abstractions during dry periods | Shepley and Streetly (2007) |

Regional hydrological features of the water management units in the two main aquifers are summarised in Table 1. Two of the water management units are situated in eastern England (Lincolnshire, unit 1) and central southern England (the Chilterns, unit 2) and are underlain by the Chalk aquifer, and two of the water management units are situated in central England (East Midlands, unit 3) and north west England (Shropshire, unit 4) and are underlain by the Permo-Triassic sandstone aquifer. The largest groundwater user in these management units is drinking water, followed by industrial water use, agricultural and environmental water use (BGS, 2015). Groundwater use is regulated using abstraction licences, which have changed since their introduction in 1963 (Ohdedar, 2017). Since the implementation of the Water Framework Directive in 2000, groundwater abstraction licences follow a water balance approach to ensure 'good groundwater status'. This resulted in a reduction of licensed groundwater abstractions (Environment Agency, 2016). Specific information regarding the change in water use in these water management units was found in previous groundwater studies (Whitehead and Lawrence, 2006; Environment Agency, 2010; Shepley and Streetly, 2007; Shepley et al., 2008). In all water management units, a dense network of groundwater monitoring sites and physically-based models are in use by water managers to observe the groundwater status at catchment scale.

## 3 Data and Methods

### 3.1 Data

The analysis has been undertaken for a 30-year period (1984-2014) using precipitation, evapotranspiration, and groundwater level time series. This time period includes at least four major groundwater droughts with national spatial extent, namely: 1988-1994, 1995-1997, 2003-2006, and 2010-2012 (Durant, 2015).

Precipitation and potential evapotranspiration data were obtained from the GEAR dataset (Tanguy et al., 2016) and the CHESS dataset (Robinson et al., 2016). The gridded (1km$^2$) GEAR dataset contains interpolated monthly precipitation estimates derived from the UK rain gauge network. The CHESS dataset is also gridded (1km$^2$) and contains climate data, from





which potential evapotranspiration estimates are computed using the Penman-Monteith equation. We aggregated daily potential evapotranspiration estimates to monthly sums for grid cells corresponding to groundwater well locations.

The precipitation estimates were converted into standardised indices (SPI) following the method of McKee et al. (1993). A gamma distribution was fit to the precipitation estimates and standardised indices were computed after converting to a normal distribution. The SPI was calculated for a number of accumulation periods (in months) in order to define the optimal correlation between standardised precipitation and groundwater time series.

     Groundwater level time series were obtained from the national groundwater database in the UK, which contains time series 120   for both reference wells and (regular) monitoring wells. 208 sites have been included in the analysis, 39 are reference sites and 170 monitoring sites. Reference wells were taken to represent near-natural conditions in the 30-year time period. These wells were selected from the Index and Observation wells listed in the UK Hydrometric Register (Marsh and Hannaford, 2008). All these Index and Observation wells have previously been assessed by the British Geological Survey and well descriptions indicate near-natural or possible (intermittent) influence of groundwater abstraction. Wells have been selected for this study 125   are categorised as being near-natural reflecting regional variation in groundwater levels with minimal abstraction impacts. This selection of reference wells includes 30 wells in the Chalk and 9 wells in the Permo-Triassic sandstone. In the four water management units, 660 monitoring sites were originally considered for the regional groundwater drought analysis. These groundwater level time series were truncated to the 30-year analysis period, after which all groundwater level observations were quality-checked. Unrealistic observations were cross-validated with available meta-data, and if unexplained, removed 130   from the dataset. Short sequences of missing data (less than 6 months) in the time series were filled using linear interpolation (Tallaksen and Van Lanen, 2004; Thomas et al., 2016). Time series with longer sequences of missing data were removed from the dataset leaving a total of 170 (out of the original 660) groundwater level time series that were deemed of good quality, of which 38 were located in Lincolnshire, 45 in Chilterns, 36 in Midlands, and 51 in Shropshire.

     Groundwater level time series from the reference wells and monitoring sites were standardised into the Standardised Ground-135   water level Index (SGI) (Bloomfield and Marchant, 2013), which is briefly explained here. Monthly groundwater observations were grouped for each calender month. The rank of each monthly observation within the 30-year time period was determined by a non-parametric fitting. The ranked observations were standardised by an inverse normal cumulative distribution to calculate the SGI value. The resulting SGI time series represent extremely low to below-normal ($-3 < SGI < 0$) and higher than normal to extremely high ($0 > SGI > 3$) monthly groundwater levels in the groundwater time series. Groundwater level ob-140   servations are physically constrained by length of the screened interval of the borehole. Therefore, the lowest SGI value might indicate that the groundwater level fell below the borehole screen and the highest SGI value can indicate the groundwater level reached the surface.

     Qualitative information about groundwater use was provided for each water management unit by the national regulator (the Environment Agency (EA) in England). Detailed maps were made available regarding the purpose and recent (dated at 2015) 145   licensed abstraction volumes (see Figure S1). In addition, reports describing the EA's regional groundwater resource models and location specific groundwater studies were used as reference material to estimate changes in groundwater use for each water management unit (Table 1).



## 3.2 Methods

The methodological framework that was developed used two approaches to investigate the impact of groundwater use on
groundwater droughts. The first approach starts with a regional near-natural groundwater drought reference based on the reference wells. The SGI time series of the reference wells are clustered to identify common spatial and temporal patterns in the near-natural groundwater levels of the two aquifers. The reference wells are taken to represent regional groundwater variation that is primarily driven by climate and hydrogeology. Then, monitoring wells in each of the four water management units were paired to these regionally-coincident clusters of reference wells (Figure 1). The occurrence and characteristics of droughts in
the monitoring wells were compared with those in the paired reference clusters to assess the potential effects of abstraction on groundwater droughts. The second approach consisted of a groundwater trend test that quantified the strength of long-term trends as a consequence of continuous impact of groundwater use in the water management units. The spatial distribution of identified trends was evaluated according to the annual abstraction licences in the water management units.

### 3.2.1 Time series clustering

Three hierarchical clustering methods: single linkage, complete linkage, and Ward's minimum were tested to find the most suitable approach for clustering the SGI time series of the reference wells. In each method, Euclidean distance was used as measure of similarity and cluster compositions that showed the least overlap between clusters were selected (Aghabozorgi et al., 2015). Criteria for the clusters were set by previous studies ( for the Chalk aquifer only) and known hydrogeological differences in the aquifers. For both aquifers, the minimum number of hydrograph clusters was sought that produced spatially
coherent clusters.

### 3.2.2 Correlation between $SPI_Q$-SGI

Under near-natural conditions, the maximum correlation between standardised precipitation and groundwater indices ($SPI_Q$-SGI) is generally high, given an optimal precipitation accumulation period (Q) that accounts for delayed recharge (Bloomfield and Marchant, 2013). In previous studies, this correlation was used to demonstrate either the absence (Haas and Birk, 2017;
Chu, 2018; Bloomfield et al., 2015; Kumar et al., 2016), or presence of human-influence on groundwater level observations (Lorenzo-Lacruz et al., 2017; Lee et al., 2018) . In this study, the presence or absence of human-influence on groundwater was determined in relation to the lowest $SPI_Q$-SGI correlation of each near-natural reference cluster. This assumes that any deviation from a correlation between the driving precipitation and the resulting groundwater level time series is primarily due to the effects of groundwater abstraction. Strongly non-linear processes in the unsaturated zone that may reduce the correlation with
groundwater levels were accounted for by using the optimal precipitation accumulation period. Based on these assumptions, the lowest $SPI_Q$-SGI correlation of the near-natural reference cluster is taken as a threshold. Monitoring wells with higher $SPI_Q$-SGI correlations are regarded as uninfluenced and those with lower correlations as potentially human-influenced. The monitoring wells are thus separated into two groups of *uninfluenced* wells and *influenced* wells. Statistical differences between the uninfluenced and influenced wells were computed using a non-parametric Wilcox test.





### 3.2.3 Drought analysis

Groundwater droughts were defined using a threshold approach applied to the SGI series. Groundwater droughts are considered to occur when the SGI value is at or below -0.84, which corresponds to a $80^{th}$ percentile as used by Yevjevich (1967), Tallaksen and Van Lanen (2004), Tallaksen et al. (2009), or a 'once every 5 year drought event'. Drought characteristics were compared between the reference and monitoring sites focusing on drought occurrence, frequency, duration, and magnitude.

### 3.2.4 Trend test

The last step of the analysis was a monotonic trend test for annual groundwater level time series. Monthly groundwater level readings were averaged using the calender year. These annual groundwater level time series were tested for monotonic trends. Our assumption was that human-influenced groundwater systems show more persistent trends compared to natural conditions. This has been shown in the literature (Thomas and Famiglietti, 2015; Sadri et al., 2016; Bhanja et al., 2017; Pathak and Do-damani, 2018), when applying a trend test to relatively short time periods, e.g. 10-30 year time series. Trends in groundwater level time series were tested using a modified Mann-Kendall trend test (Mann, 1945; Kendall, 1948), which includes a modification developed by Yue and Wang (2004) to account for significant autocorrelation in the annual groundwater data (Hamed, 2008). The trend Z statistics (Z) indicated increasing (Z>0) or decreasing (Z<0) trend direction. Z values over |2| were considered significant (Panda et al., 2007). Climate time series (P and PET) were also tested for trends using annual data to compare groundwater trends with trends in climate data. Trends in the non-autocorrelated climate data were tested using the standard Mann-Kendall trend test.

## 4 Results

### 4.1 Near-natural groundwater reference clusters

The near-natural reference clusters, based on the clustering of SGI time series of the reference wells and the clustering criteria, were defined by Ward's minimum clustering technique. The Ward's minimum cluster composition shows the least overlap between clusters of the three clustering techniques that were used (Figure S2). Eight clusters are identified, of which five clusters are located in the Chalk (C1-5) and three in the Permo-Triassic Sandstone (S1-3) (Figure 1). The spatial distribution of Chalk clusters (C1, C3, C4) is consistent with clusters previously identified by Marchant and Bloomfield (2018). A separate cluster is identified in East Anglia for 5 reference wells (C2). The smallest cluster is C5 (2 wells), for which the cluster dendrogram shows a small difference in similarity between C4 and the 2 reference wells in C5 that are located close to the coastline (cluster dendrogram result not shown; difference between C4 and C5 is shown in Fig. S2). C1 and C3 are coincident with water management unit 1 and 2, and are used as near-natural reference for monitoring sites in those units. In the Permo-Triassic sandstone aquifer, only one spatially coherent cluster (S2) is found when all nine SGI time series are clustered (Figure 1). The cluster composition of the other two smaller clusters (S1 and S3) is not spatially coherent and there is no evidence of





previous clustering studies available that can confirm these two clusters. Hence, only S2 is used as near-natural reference for monitoring sites in water management units 3 and 4.

The maximum $SPI_Q$-SGI correlations of the reference wells vary between 0.66 and 0.89. These correlations are found using the optimal accumulation period, which accounts for delay in recharge that is different for each reference cluster. C1 represents a relatively fast-responding section of the Chalk and has a short Q of 12.6±5.4 months. The Q of C2 and C3 is

higher, respectively 24±8.6 and 18.2±4 months. This corresponds to the delay in groundwater recharge due to the Quaternary deposits present in these regions (Allen et al., 1997). In the South East, the Chalk is highly fractured, which is reflected by a short Q of 8±2.1 months for C4 and C5. In the Permo-Triassic sandstone, the Q of S2 is 35±4.5 months, which confirms a slow-responding groundwater system (Allen et al., 1997).

In the monitoring sites, the majority of the $SPI_Q$-SGI correlations are as high or higher than the minimum correlation

of paired reference clusters. These monitoring sites are therefore considered uninfluenced by abstraction and the range in optimal correlations between them is most likely related to local hydrogeological settings (e.g. aquifer depth and semi-confined sections). The accumulation periods for the monitoring sites within the management units is variable and appears to be in part a function of aquifer depth and the local nature of aquifer confinement (Figure S3). For example, shorter accumulation periods are found in shallow sections of the aquifer (East Shropshire and West Chilterns), and in outcrops (East Lincolnshire).

Longer accumulation periods are found in deep sections of the Permo-Triassic aquifer (West Shropshire) and semi-confined sections of the Permo-Triassic (Midlands) and Chalk aquifer (East Chilterns, and South East Lincolnshire). The percentage of uninfluenced sites varies between the water management units. The largest percentage is found in the Chilterns (71%), followed by the Midlands (63%), Shropshire (53%), and Lincolnshire (31%). Monitoring sites with a $SPI_Q$-SGI correlation below the minimum correlation of the paired reference cluster are treated as possibly influenced by abstraction.

## 4.2 Groundwater droughts

Groundwater droughts observed in the reference clusters show variation due to spatial patterns in both precipitation and hydrogeology. The NW-SE precipitation gradient in England results in different precipitation patterns, which might be reflected in the variation in groundwater drought occurrence (Figure 1). For example, in C1 groundwater levels are low in 2003-06, but not below the drought threshold. In C2, groundwater levels are slightly lower and a short drought event is observed in the

SGI cluster mean. In C3-5 and S2, however, the 2003-06 drought event was a major drought event. Spatial variation in the hydrogeology results in varying drought duration for the Chalk clusters. In central England, longer drought durations are found in clusters C2 and C3. This region is partly covered by Quaternary deposits, which delays recharge and prolongs droughts for the reference wells. Shorter (and more frequent) events are observed in C4 and C5, which are located in highly fractured Chalk.

The average drought characteristics (duration in months, magnitude in accumulated SGI over the drought period, and fre-

quency) for monitoring sites in each water management unit show differences between uninfluenced and influenced sites, see Table 2. Shorter and less intense, but more frequent drought events are observed in the influenced sites in Lincolnshire, Chilterns, and Shropshire. In these water management units, the difference in drought duration and frequency is significant. Droughts are observed twice as often in the influenced compared to the uninfluenced sites in Lincolnshire and Chilterns, but







**Figure 1.** Eight clusters based on the 39 reference groundwater sites in the Permo-Triassic sandstone and Chalk aquifer are shown, representing long-term near-natural groundwater level variation. All time series are standardised for the 30-year time period (1984-2014). In the centre, locations of the reference wells are shown marked by the dots in different colours for all eight clusters. The four water management units are indicated in dark red (groundwater monitoring sites in triangles). Three of these units coincide with reference clusters: 1:Lincolnshire (C1), 2: Chilterns (C3), and 4: Shropshire (S2). S2 is also used to compare water management unit 3 (Midlands) as this is the nearest reference cluster in the Permo-Triassic sandstone. In the panels left (Permo-Triassic sandstone) and right (Chalk), SGI time series are shown for each cluster, showing the cluster mean (thick line), the range of all reference wells in the cluster (shaded coloured area) and reference droughts of the cluster mean (filled area).





**Table 2.** Average drought characteristics: duration, magnitude, and frequency of all monitoring sites in the four water management units. The monitoring sites are separated using the lower limit of the cluster SPI$_Q$-SGI into *uninfluenced* and *influenced*. Differences between the two groups are tested for significance using a Wilcox test. Tests for which the p<0.05 are in **bold**.

|  | Uninfluenced wells (%) | Average duration (in months) | | Average magnitude (from SGI) | | Average frequency | |
|---|---|---|---|---|---|---|---|
|  |  | Uninfluenced | Influenced | Uninfluenced | Influenced | Uninfluenced | Influenced |
| 1: Lincolnshire | 31 | **7.6** | **3.3** | -3.4 | -1.5 | **11.0** | **24.9** |
| 2: Chilterns | 71 | **8.67** | **3.4** | **-3.9** | **-1.54** | **10.0** | **25.4** |
| 3: Midlands | 63 | 9.89 | 11.6 | -4.5 | -5.3 | **9.5** | **9.0** |
| 4: Shropshire | 53 | **6.8** | **5.0** | -3.1 | -2.3 | **11.9** | **15.7** |

this difference is smaller in Shropshire. In the Midlands, the average drought duration of influenced sites exceeds the drought
duration in uninfluenced sites. Longer and more intense groundwater droughts occur less often in influenced sites, which is in contrast with the other water management units. However, only the difference in frequency is statistically significant.

The drought characteristics in Table 2 suggest that drought events vary widely within and between water management units. These differences are shown in a combined time series plot in Fig. 2. For each water management unit, there are two plots. The upper plot shows the SGI hydrograph of the reference cluster with the cluster mean and drought events highlighted. The
lower plot shows periods of drought colour-coded by the drought intensity at individual monitoring sites. The monitoring sites are sorted from high to low SPI$_Q$-SGI correlation. The cluster minimum SPI$_Q$-SGI correlation is indicated with a dashed line, i.e. 0.75 for Lincolnshire, 0.71 in the Chilterns, and 0.69 in the Midlands and Shropshire. Figure 2 shows that the timing of droughts in uninfluenced wells aligns mostly with droughts of reference clusters. In Lincolnshire and Chilterns, sites with a SPI$_Q$-SGI correlation higher than the cluster minimum (uninfluenced sites) had drought events similar to the reference sites.
Influenced sites (those with SPI$_Q$-SGI correlations lower than the cluster minimum) had typically shorter drought events of a lower magnitude. In Shropshire, additional droughts are found before and after drought events in the reference wells. However, these additional events are not exclusively observed in influenced sites. In nearly all monitoring sites, additional drought events are found in 1984, 1989-90, 1995-96, 2005-06, and 2009, which is prior to a long drought event for all cases, except for 1984. Contrastingly, longer and more intense droughts are observed in all Midland sites in 1990-95. Droughts observed in influenced
sites are also longer in 1984-1986, 1997-2001, and 2005-06 compared to the reference cluster and fewer droughts are observed in 2010-12.

The additional events in influenced sites coincide with low SGI values in the reference wells that sometimes occur prior to a long drought event. For example, additional droughts are observed in 1984, 1995-96, 2005-06, and 2014 in Lincolnshire, and in 1984-86, 2004, and 2009-10 in the Chilterns. In those periods in both water management units, the reference cluster
mean was below 0, but not below the drought threshold. In the case of 1995-96, 2004, and 2009-10, these additional drought events occurred prior to a long drought event. It could be that a sudden increase of groundwater use pushes groundwater level below the drought threshold in influenced sites. Drought descriptions in the literature show an increase in water demand during the 1995-97, 2003-06 and 2010-12 drought (Walker and Smithers, 1998; Marsh et al., 2013; Durant, 2015). Hot summers,





heatwaves or dry conditions can increase the local groundwater use. Another explanation for increased groundwater use could

be related to surface water use restrictions (voluntarily or mandatory) that might be in place before a major groundwater drought (Rey et al., 2017; Rio et al., 2018). The reduced surface water availability is then replaced with groundwater, resulting in lowered groundwater levels and potentially aggravating a groundwater drought.

Overall drought magnitude seems to be decreasing since the 1995-1997 drought event. The droughts observed in 2003-2006 and 2010-12 are shorter and of lower magnitude than the 1995-97 drought in most sites. This is seen most convincingly

in Lincolnshire, Chilterns and the Midlands, where the magnitude of droughts decreases dramatically over the 30-year time period. In Shropshire, this tendency is less strong, as the 2010-12 drought was of a similar magnitude as the 1995-1997 drought.

## 4.3 Trends in groundwater

Significant trends in groundwater level have been detected in 48% of all monitoring wells in the water management units. The strength of these significant ($Z > |2|$) trends varies between and within management units, and both upward and downward

trends have been identified (Figure 3). Overall, 27% of the significant trends are upward (positive), indicating a sustained rise in the 30-year groundwater level time series, compared to 21% of significant downward (negative) trends that indicate sustained lowering of groundwater levels. The presence of these significant trends is notable given the weak, non-significant, trends in the 30-year precipitation and potential evapotranspiration data (P: $Z = -0.75$ - 1.53, PET: 0 - 0.65).

The direction of trends in groundwater and their spatial coherence within the water management units show different pat-

terns. In the Chalk water management units, positive trends dominate. In Lincolnshire, 9 out of the total 25 positive trends are significant, compared to 9 out of 32 in Chilterns. There are fewer sites with a significant negative trend in both water management units. This is respectively 4 out of 11 in Lincolnshire and 4 out of 12 in Chilterns. In Lincolnshire, sites with a negative trend are, all but one, located in the semi-confined Chalk. This is in sharp contrast with the semi-confined Chalk in Chilterns, where mainly (significant) positive trends are found. In the Permo-Triassic sandstone, the majority of monitoring sites have a

significant trend (69% in Midlands and 53% in Shropshire). In the Midlands, more positive than negative trends are detected. In total, 18 out of 25 positive trends are significant, compared to 7 out of 11 significant negative trends. Positive trends are mainly found in the centre of the water management unit. Negative trends are found north and south of that. In Shropshire, more negative than positive trends are detected. 31 sites have a negative trend, of which 17 significant. These trends are mainly detected in the west of the water management unit. Positive trends are mainly located east in between two fault lines (Oller-

ton and Childs Ercall Fault (Voyce, 2008)). Half of these positive trends (20 in total) are significant. In Fig. 3, the maximum licensed abstraction volume is also shown. These licences show in which aquifer sections groundwater is primarily abstracted. However, without a record of the actual use of these licences it is impossible to directly relate the detected trends to these abstraction locations.







**Figure 2.** Drought occurrence visualised for all four water management units: 1:Lincolnshire, 2:Chilterns, 3:Midlands and 4:Shropshire. The top panel shows the SGI hydrograph of the reference cluster mean based on reference wells. The range of the reference cluster is coloured in grey. The dotted line represents the drought threshold for the cluster mean with shaded areas for the reference drought events. These reference drought events are also shown in long grey panels in the lower plot that shows the individual droughts as found in monitoring sites in each water management unit. The length of coloured bars indicates the drought duration, whereas the colour represents the drought magnitude of each drought in blue-red scale for accumulated SGI.





**Figure 3.** Trend values for monitoring wells in the four water management units (1: Lincolnshire, 2: Chilterns, 3: Midlands, 4: Shropshire). The red and blue diamonds indicate the positive or negative Z values for the Modified Mann-Kendall trend test for each monitoring well. Z values over |2| indicate a significant trend in the 30-year groundwater time series.



## 5 Discussion

The presented results of the two main aquifers in the UK show that groundwater droughts in the Chalk and Permo-Triassic sandstone aquifer are primarily driven by precipitation and modified by the hydrogeology setting and groundwater use. The precipitation gradient was the primary driver for regional variation in near-natural groundwater droughts in 1989-1992 and 2003-06, which is confirmed by the work of Bryant et al. (1994) and Marsh et al. (2007). This explains the absence of a groundwater drought in the 2003-06 period in the northern Chalk (C1), compared to the southern Chalk (C2-C5). Regional

variation of near-natural droughts within the different hydrogeological units was linked to the hydrogeological setting, as accumulation period varied in each reference cluster. These accumulation periods align with previous findings of Bloomfield and Marchant (2013). On a smaller scale, accumulation periods varied gradually within the water management units, as a function of aquifer depth and confinement of the aquifer, which was also found by Kumar et al. (2016), Van Loon et al. (2017) and Haas and Birk (2017). The relation between accumulation period and groundwater drought duration, as observed

in the reference clusters, corresponds to the relation between groundwater memory and drought duration for near-natural observations, as found by Bloomfield and Marchant (2013).

Influence of groundwater use on groundwater droughts is detected in a subset of monitoring sites in each of the four water management units. This subset often represents a minority of monitoring sites in the water management unit. Two patterns are found in the water management units that illustrate an asymmetric impact of water use on groundwater droughts. The first

pattern (found in three water management units) is that of more, but shorter and less intense droughts that are observed in the influenced compared to uninfluenced sites. The second pattern (found in one water management unit) shows the opposite impact with less, but longer groundwater droughts in influenced compared to uninfluenced sites. Both patterns are inferred as a direct consequence of groundwater use in the water management units.

The first pattern, apparent in Lincolnshire, Chilterns, and Shropshire, shows an increase in short drought events often found

before a major drought event or during hot summers, which is probably related to an increase in water use (Walker and Smithers, 1998; Marsh et al., 2013; Durant, 2015) and/or complementary groundwater use due to surface water use restrictions (Rey et al., 2017; Rio et al., 2018). We see the effect of this local increase in water use in our data in the temporarily lowered groundwater levels, resulting in additional drought events. The short duration and low intensity of these additional droughts suggests that local groundwater levels recover quickly. Whether groundwater was removed from groundwater storage or capture

(impacting environmental flows) remains unknown (Konikow and Leake, 2014), although the short duration and rapid recovery suggest that an equilibrium was established soon after the abstractions. Regional groundwater model studies in these three water management units show that the annual average *actual* abstractions are smaller than modelled recharge for Lincolnshire, Chilterns, and Shropshire. The ratio abstraction to recharge is 0.67 (Hutchinson et al., 2012), 0.5 (Environment Agency, 2010), 0.5 (Shepley and Streetly, 2007) for the three water management units respectively. Even though these ratios are calculated

using data from different regional groundwater models, the results show that the long-term balance between groundwater use and recharge is positive, which might be the reason that the overall influence of abstraction on groundwater droughts is relatively minor with a reduced drought intensity and duration for influenced sites.





The second pattern, apparent in the Midlands, shows intensified groundwater droughts that occur less often. Most of the intense drought events are observed prior to 2001 with lengthened droughts in 1984-1986, 1990-95, 1997-2001. Lengthening

of droughts is a common phenomenon in overused groundwater systems (Custodio, 2002). In the Midlands, prior to 2000, groundwater abstraction exceeded the modelled recharge by 25% (Shepley et al., 2008). The overabstraction resulted in lower streamflows in the area (Shepley et al., 2008), suggesting that the balance between water removed from capture and storage was disrupted (Konikow and Leake, 2014). Reforms of water allocations in 2000 have reduced groundwater abstractions to meet the long-term water balance. These long-term changes in groundwater abstractions match with the majority of significant

positive groundwater trends in the Midlands.

The long-term influence of groundwater use is inferred from identified trends in the groundwater time series. Large spatial differences are found in the strength and direction of groundwater trends in both aquifers, whilst trends in precipitation and potential evapotranspiration are negligible. Positive groundwater trends dominate, which may be a result of overall rising groundwater levels due to a reduction of groundwater use since 1984 (start of the investigation period of this study). A gradual

or immediate reduction of water use can restore the balance between groundwater use and recharge (Gleeson et al., 2010; Konikow, 2011), although it can take decades before an equilibrium is reached (Gleeson et al., 2012). Overall, groundwater droughts show a reduction in magnitude and duration from 1984 to 2014. Most intense droughts are found during in the first two decades (1984-2004) of the time period. Even though this coincides with a reduction of groundwater use, more research is required to distinguish the climate-driven droughts from the human-modified droughts.

A conceptual typology is presented in Figure 4 summarising near-natural drought, two types of human-modified droughts as found in the water management units, and an extreme condition of human-modified drought. Under near-natural conditions, groundwater droughts occur given the climate forcing and hydrogeological setting (upper panel in Figure 4). Under human-influenced conditions, the impact of groundwater use on groundwater droughts is asymmetric. In regions where the annual average groundwater use is smaller than the annual average recharge, the frequency of groundwater droughts increases, result-

ing in shorter events of a lower magnitude (second panel in Figure 4). This corresponds to the 'dynamic sustainable range' as presented in the conceptual model of Gleeson et al. (2019). In regions where the annual average groundwater use approaches annual average recharge, the opposite is found with less, but prolonged droughts of higher magnitude and duration (third panel in Figure 4) corresponding to strategic aquifer depletion, when meeting the dynamic sustainable range over a long time scale (Gleeson et al., 2019). The last panel shows the extreme conditions of groundwater depletion, in which groundwater droughts

are not recovering by the average annual recharge and groundwater levels tend to fall consistently. These extremes conditions are not identified in the UK, but heavily intensified and lengthened droughts are found in California (He et al., 2017), Australia (Leblanc et al., 2009), Spain (Van Loon and Van Lanen, 2013), Bangladesh (Mustafa et al., 2017) and India (Asoka et al., 2017).

Further research is required to analyse the effects of water use changing over time to groundwater droughts. In this study,

we have investigated the overall long-term impact of groundwater use using monotonic trends in groundwater. A different methodology is required to evaluate the impact of new water regulations on groundwater droughts (Bhanja et al., 2017). For example, an observation-modelling or conceptual modelling approach can be used to differentiate pre- and post-regulation



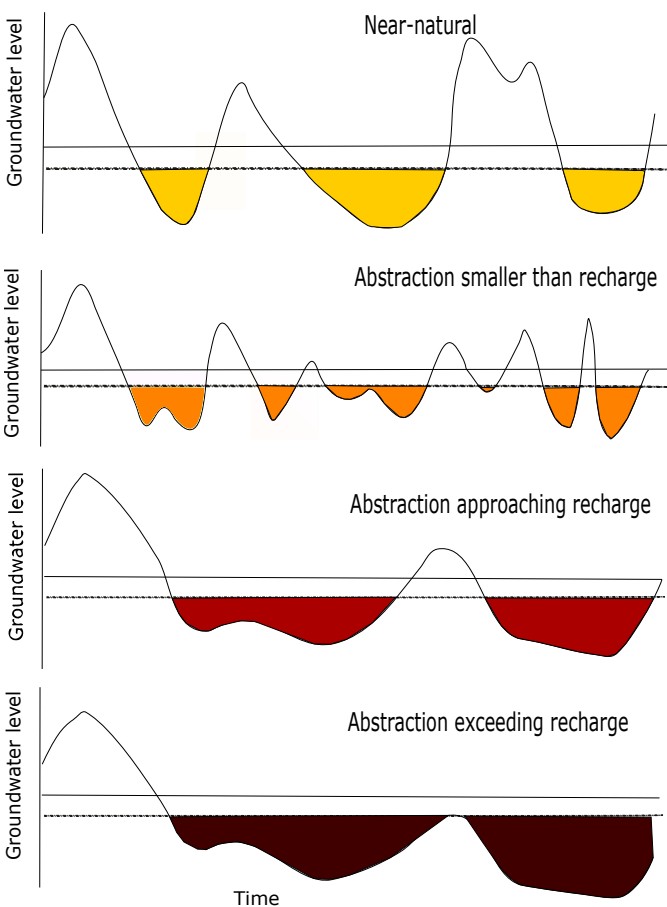

**Figure 4.** Conceptual figure summarising the near-natural groundwater droughts, two identified patterns in the four water management units, and one extreme scenario of groundwater depletion. The top panel shows groundwater droughts under near-natural conditions, the lower three panels show groundwater droughts under human-influenced conditions with increasing intensity of impact of groundwater use. The second panel shows typical groundwater droughts when annual average abstractions are smaller than the annual average groundwater recharge. The third panel illustrates groundwater droughts when annual average abstractions approaches recharge, and the last panel shows extreme groundwater drought conditions when average annual abstractions exceed recharge.





groundwater droughts (Van Loon et al., 2016b; Kakaei et al., 2019; Liu et al., 2016). This future modelling work could also provide long-term context for water management effects, natural variability, non-stationary effects of anthropogenic climate

change (specifically warming) on changes in groundwater drought characteristics (Bloomfield et al., 2019).

Further applications of this study could be beneficial for water regulators and scientists alike, as the presented conceptual typology can be used to investigate the impact of groundwater use without having to obtain time series of actual groundwater abstractions. The developed methodology shows how qualitative information on groundwater use and annual long-term averages aid to get a better understanding of asymmetric impact of groundwater use on groundwater droughts. Considering the

large-scale modification of the hydrological cycle and the consequences for droughts (Van Loon et al., 2016a), it is important to further this approach and investigate the sustainable use of groundwater resources (Gleeson et al., 2019).

## 6   Conclusions

The impact of groundwater use on groundwater droughts is investigated based on a comparison of potentially influenced groundwater monitoring sites and near-natural, or largely uninfluenced reference sites. Results show that long-term groundwa-

ter use has an asymmetric impact on groundwater droughts for a subset of influenced groundwater sites. A conceptual typology summarises these different patterns in groundwater drought occurrence, duration, and magnitude. The first type shows an increase in groundwater droughts with a low magnitude, of which the timing coincides with periods of a high water demand, for instance during heatwaves. This is found in three water management units where the long-term water balance is positive and annual average groundwater abstractions are less than the recharge. The second type is marked by lengthened, more intense

groundwater droughts. This is found in one water management unit where annual average groundwater abstractions temporarily exceeded recharge. The balance between long-term groundwater use and recharge seems to explain the asymmetric impact of groundwater use on groundwater droughts. However, more research is required to investigate the impact of changes in water use. During the period of investigation, regulated groundwater abstractions have reduced and our results show a majority of rising groundwater trends based on 30 years of data. Further research could potentially indicate how droughts are affected by

these changes in water use.

In conclusion, this study presents a conceptual typology to analyse groundwater droughts under human-modified conditions. We found that human-modified droughts differ in frequency, magnitude, and duration dependent on the long-term proportional groundwater use compared to recharge. This highlights the relation between long-term and short-term groundwater sustainability.





*Code availability.*    The code is available upon request.

*Data availability.*    The raw groundwater time series and abstraction locations can be obtained via the Environment Agency. Standardised
groundwater level time series is available upon request.

*Author contributions.*    DW, AVL, BP, DH conceived and designed the study. DW performed the analysis and wrote the paper, supervised by
AVL, BP, DH. All authors contributed to the manuscript.

*Competing interests.*    The authors declare no conflict of interest.

*Acknowledgements.*    We would like to thank Richard Morgan and Catriona Finch for providing data and their valuable feedback in the
initial stages of this study, and Michael Kehinde, Vicky Fry, Alex Chambers, and Kevin Voyce for providing groundwater monitoring
data and background material. The study has benefited from valuable discussions during meetings and workshops of the 'Drought in the
Anthropocene' working group of the IAHS Panta Rhei network, and we would like to thank Henny Van Lanen in particular. Financial support
for DW was provided by a CENTA NERC grant (NE/lL002493/1) and CASE studentship of British Geological Survey (GA/16S/023). JPB
publishes with permission of the Director, British Geological Survey (NERC, UKRI).





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
