# Peer review of "Asymmetric impact of groundwater use on groundwater droughts"

_Hydrology and Earth System Sciences, 2020_

## Referee Comment (RC1) · Anonymous Referee #1 · 12 Mar 2020

This study uses a framework that consists of two approaches, and conducted a case study in UK to investigate the impact of groundwater use on groundwater droughts. Generally, the manuscript is well organized with clear logic, before I recommend it for publication, major improvements are still needed, particularly for the method they used for recognizing the presence or absence of human-influence on groundwater. Please find my specific comments below:

1.Lines 171-172: 'In this study, the presence or absence of human-influence on groundwater was determined in relation to the lowest SPIQ-SGI correlation of each near-natural reference cluster'. I think it is questionable to determine the presence or absence of human influence depending on the correlation analysis. For example, for a certain site, SGI is best correlated to SPI at short time scales. Due to human interference, the drought duration indicated SGI may become longer, leading to SGI best correlated with SPI at longer time scales. The increased time scale of SGI does not necessarily corresponds to reduced correlation of SPIQ-SGI, and the correlation may also increase. Moreover, considering the significant spatial heterogeneity of groundwater (e.g., groundwater of the monitoring sites may show different patterns from reference sites ), it would be better to recognize human influences by analyzing the temporal variation of groundwater for the same site (e.g., compare the statistics of groundwater among different decades). The uncertainty derived from the method for recognizing the presence or absence of human-influence on groundwater needs to be discussed.

2.Section 3.2.4 Lines 185-196. The authors use the statistic variable 'Z' of the Mann-Kendall test to judge whether the groundwater of the monitoring sites involves human influences. I think the statistic variable 'Z' can indicate the significance level (e.g., when |Z|>2.56, it suggests a significant trend), however, it seems arbitrary to conclude that the detected trend becomes more significant with increased value of |Z|. Fortunately the authors mentioned that both PET and precipitation present no significant trend, while groundwater level presents significant trend. This inconsistent pattern between PET /precipitation and groundwater level may imply the existence of human influence. I suggest the authors use additional methods (e.g., linear regression) to confirm the existence of human influence.

3.The time series of SGI for reference wells in Figure 1 (Section 4.1) show significant spatial heterogeneity, and their time scales vary from one site to another. For example, C2 #5 presents long time scales, while C4#9 presents short time scales. This may lead to the higher correlation between SPIQ-SGI for C2 #5 than C4#9 (see comment 1). I think the way of using correlation to judge the human influence is worth thinking.

4.Lines 254-264 and 315-318: The authors mentioned that 'The first pattern, apparent in Lincolnshire, Chilterns, and Shropshire, shows an increase in short drought events often found before a major drought event or during hot summers, which is probably related to an increase in water use'. However, from Fig.2 it shows that 'minor droughts

before major drought events' are not limited to influenced sites, similar phenomena are also observed in uninfluenced sites. Other factors such as the drought identification method, and the spatial heterogeneity of groundwater may also generate such minor droughts. It seems arbitrary to attribute such events to the increased water use and there is much uncertainty on the results.

5.Lines 322-323, The authors mentioned 'We see the effect of this local increase in water use in our data in the temporarily lowered groundwater levels, resulting in additional drought events'. Could you provide additional information on the evolution process of water use and droughts, e.g., show the time series of both water use data and groundwater levels in one figure.

6.Lines 115-118ïïjŽThe authors failed to illustrate how they calculate SGI clearly. For example, which probability distribution was employed to fit the groundwater series. Whether the impact of data seasonality was considered when calculating SGI? More details on the computation of SGI should be added.

7.Lines 120-121: '208 sites have been included in the analysis, 39 are reference sites and 170 monitoring sites. ' Here '208 sites' should be '209 sites (170+39=209)'.

8.Lines 130-131: How do you fill the missing sequences, using the time series of adjacent sites? Details on the linear interpolation method should be supplemented. Besides, sites with missing data more than 6 months would be removed directly?

9.The current form of Fig. 2 makes it difficult to judge the impact of human influences. The authors could add the time series of SGI for the monitoring sites so that readers can easily find human influenced periods.

---

## Referee Comment (RC2) · Anonymous Referee #2 · 14 Apr 2020

This paper investigates the impact of groundwater use on groundwater drought for a case study in the UK.

Overall, I found the paper to be well-written, with some interesting results supported by some nice figures. The work represents a useful contribution to better understand how groundwater use affects groundwater drought and how groundwater levels have changed over time in the UK.

My main suggestions for the paper are to improve the clarity of the methods and reflection of the variability in groundwater levels. I agree with reviewer 1 that there is a lot of uncertainty in the results and some of the links between groundwater use and groundwater drought are somewhat arbitrary. I sympathise with the authors as I know how difficult it is to gain groundwater use data that may help make these findings more
robust, nevertheless, I believe the authors could make more efforts to discuss the limitations of their results and report the uncertainty/variability in their results.

Please see more detailed comments below:

Main Comments

Calculation of SPI and comparison with SGI -

In the methods/discussion please add some comment on the choice of gamma distribution used to calculate SPI. Other studies have shown that this is often not the most appropriate distribution for precipitation data and it would be good to discuss the impacts of this (see Svensson et al. 2017 for example).

From the methods section, it seems that you compare the SPI from a single grid cell with the corresponding groundwater well location (this should be clarified in the text). It would be good to add to the discussion the impacts of comparing a 1km2 grid cell of SPI with SGI that is a product of a regional groundwater aquifer system and regional rainfall patterns.

Methods and terminology

The methods (in places) were not clear – in particular, the SPI\_SGI correlations and the use of the near-natural wells, uninfluenced and influenced monitoring sites. It would be useful to have a worked example of how the SPI-SGI correlations work in practice (showing an example for two sites – one influenced and one non-influenced and how they compare to the near natural reference cluster). It would also be useful to have a map of the influenced and non-influenced wells (this is maybe already included in Figure 1 but this figure is quite busy so it is hard to tell!) – are there any spatial patterns?

Reflection of uncertainty/variation

There is a lot of variation in the groundwater levels between sites and this needs to be
better reflected in the results. I suggest that the authors report the min/max or 5th/95th percentile of their results alongside the average in Table 2 and elsewhere in the text.

Like Reviewer 1, I am somewhat sceptical of attributing the shorter droughts in Lincolnshire, Shropshire and the Chilterns to water use and/or hotter Summers. Firstly the years that were identified in L263-265 did not have particularly hot summers (or this is certainly not consistent for these years) and many of these drought events can also be identified in the uninfluenced wells. These uncertainties need to be reflected in the discussion or the methods for identification need to be more robust.

**Minor Comments**

1. The abstract is quite long – I would shorten it and just highlight the key results. Currently, your more interesting results get a little lost in all the text.

2. P3 L80 It would be good to name these four water management units in the text

3. P5 L117 – What accumulation periods did you calculate SPI over, you need to be more specific here.

4. Table 1 – what time period were the long term precipitation and PET calculated over? It would be good if this was consistent with the time periods used in your study.

5. P7 L194 – Were these the climate time series from a single grid cell?

6. Table 2 - I was a little surprised that the average drought frequency for the Midlands cluster is significant when the values are quite similar (9.5 for uninfluenced and 9 for influenced) – is this correct?

7. Section 4.3 - in this section you don't distinguish between 'influenced' and 'uninfluenced' wells. It would be useful know whether the strong trends are just in the 'influenced' wells? If they are not, then your 'uninfluenced' wells may be more affected than suggested.

References Svensson, C., Hannaford, J., and Prosdocimi, I. (2017), Sta-
tistical distributions for monthly aggregations of precipitation and streamflow in drought indicator applications, Water Resour. Res., 53, 999–1018, https://doi.org/10.1002/2016WR019276.

---

## Author Comment (AC1) · 15 Apr 2020

**General response to Reviewer 1**

We would like to thank Reviewer 1 for their careful reading of the manuscript and their constructive comments. In this general reply, we would like to respond to comments 1-3 to allow further discussion. The other comments (4 - 9) will be addressed later in the new version of the manuscript, because some comments (4, 5, and 9) involve gathering additional material.

Comment 1

The first comment of Reviewer 1 deals with our use of correlation analysis to determine the human influence on groundwater level time series. We understand the concern raised in comment 1 and recognize that much of this can and will be addressed by changes to the framing and phrasing of the statements at Lines 171-172. We also agree with the reviewer that we need to include additional discussion of uncertainties associated from the method for recognizing the presence or absence of human-influence on groundwater. This discussion will be included in revisions to the text in Section 3.2.2 and will draw on our comments below.

Reviewer 1 questions the use of correlation analysis to determine the presence or absence of human influence on groundwater drought and observes that, for example, anthropogenic influences on groundwater drought status might increase $SPI_Q$-SGI correlations at longer accumulation periods. If this is the case such relatively high correlations between $SPI_Q$-SGI in the current scheme might not represent sites unaffected by anthropogenic activities (e.g. abstraction) during droughts. Finally, they suggest that analysing temporal variations in groundwater may better help recognize human influences.

However, we would observe that there are four main reasons why we believe that our approach is appropriate, as follows: 1) the definition and nature of SGI and SPI and the long-term average nature of $SPI_Q$-SGI correlations, 2) the high $SPI_Q$-SGI correlation of near-natural reference clusters, 3) the irregular and dynamic nature of groundwater abstraction in the water management units, and 4) consistency with the results of previous studies.

We would like to emphasize that correlations between standardized precipitation and groundwater time series are generally high in unconfined systems and for near-natural conditions (Bloomfield & Marchant, 2013; Bloomfield et al., 2015).. SGI and SPI are estimated for a continuous period that includes all seasons and both anomalously dry

and wet periods. So relatively high SPI$_Q$-SGI correlations are associated with near-natural conditions and represent a long-term average relationship. Under these near-natural conditions, anomalies in precipitation propagate with a relatively constant delay in recharge to the groundwater. This is due to, subsurface controls on recharge, the antecedent condition of the land surface, and non-linear response of groundwater systems (Peters et al., 2006; Tallaksen et al., 2009; Eltahir and Yeh, 1999). This constant delay is included in the correlation analysis, as the optimal precipitation accumulation period is selected when calculating the SPI$_Q$-SGI correlation.

There are two main reasons why the long term average, high SPI$_Q$-SGI correlations may be reduced. The first reason for reduced long-term SPI$_Q$-SGI correlation is when groundwater level response becomes disconnected from driving precipitation under confined conditions (Bloomfield, et al. 2015; Lee, et al. 2018). This is not considered to be a significant issue with the sites that we have investigated, as only a few sites are semi-confined (see notes in Table 1). We will modify the section 3.2.2 to make this point. The second reason for reduced long-term SPI$_Q$-SGI correlation is the effects of abstraction. In this study, groundwater abstraction is conceptualised as exerting change in groundwater storage and hence groundwater levels, independent of natural changes in groundwater storage associated with changes in precipitation. We don't have quantitative information about either the detailed operational practices during individual episodes of drought or the long-term changes in abstraction and management practices in the study areas. However, we have sufficient evidence that both are likely to have changed in an *ad hoc* and potentially irregular manner. Consequently, our working hypothesis is that where either or both occur this will contribute to a reduction in the long-term average SPI$_Q$-SGI correlation.

Groundwater abstractions in water management units (i.e. a well field) are likely to vary in space and time, as multiple abstraction wells are used to meet the water demand. The amount of abstracted groundwater depends on variable groundwater demand, management policies in place, and practical local constrains for groundwater
abstraction. For example, water demand is often seasonal with higher abstraction in spring and summer. This seasonal change in water use was previously found to reduce correlations (Lorenzo-Lacruz, et al. 2017). At the national scale and over the longer-term, we know that groundwater abstraction in England increased up until the late 1980s since when legislation has resulted in a general reduction in groundwater abstraction, but with a redistribution of where water is taken from to minimise the impacts of surface flows (Ohdedar, 2017; Whitehead and Lawrence, 2006; Environment Agency, 2010; Shepley and Streetly, 2007; Shepley et al., 2008).

Our conceptual model is that this highly dynamic pattern of groundwater abstraction, variable in space and time, will result into reduced $SPI_Q$-SGI correlations between precipitation and groundwater time series that are ordinarily seen in unconfined natural systems. This was also concluded by Bloomfield et al. (2015), who found lower $SPI_Q$-SGI correlations for wells that are influenced by groundwater abstractions (clusters 3 and 6). Another example of disturbance of this relationship is given by Haas et al. (2017), who showed that correlations between precipitation, streamflow, and groundwater observations are reduced due to the interference of power plants.

Reviewer 1 has suggested that anthropogenic influences on groundwater drought status might increase $SPI_Q$-SGI correlations at longer accumulation periods. However, given the long-term average nature of the correlation statistic this would only occur if sustained abstraction effects were felt for the majority of the period, not just for the periods of drought. However, we have no evidence that this has occurred at any of the sites in any of the study regions. The complexity and irregularity of management practices across the study sites combined with the lack of quantitative information on abstractions have also mitigated against our use of an analysis of temporal variations in groundwater response to abstraction.
Comment 2

In comment 2, Reviewer 1 questions our use of the 'Z' statistic of the Mann-Kendall test and suggests to apply linear regression instead. We agree with the reviewer that the description of the trend Z indicator [R 191-194] could be improved. The 'Z' statistic of the non-parametric Mann-Kendall trend test indicates indeed the significance level of a mono-tonic trend and the significance level does not directly indicate the impact of groundwater use. We will improve the manuscript by clarifying the interpretation of the Mann-Kendall trend test. We intended to show that the 'Z' statistic of the modified Mann-Kendall trend test indicated how much the trends deviate from the null hypothesis (no trend). Given the absence of trends in the precipitation and evapotranspiration time series, we assumed that trends in groundwater time series are related to the change in groundwater abstraction.

The significant auto- and serial correlation in the groundwater time series limits the application of parametric trend tests, such as linear regression, which is only applicable to normally distributed, independent data. We tested the groundwater time series and only 5 out of 170 time series are normally distributed (Shapiro-Wilk Normality Test). All others (165 time series) deviate from a normal distribution, which was also found by Bloomfield & Marchant (2013), for their groundwater time series. Therefore it seems unsuitable to apply linear regression to the majority of the groundwater dataset.

Comment 3

In comment 3, Reviewer 1 refers to the spatial heterogeneity shown in Figure 1. In this Figure, 8 near-natural groundwater clusters are shown and their droughts are highlighted in the SGI time series. Reviewer 1 is concerned that the difference in spatial heterogeneity would result in a higher $SPI_Q$-SGI correlation in case of longer accumulation periods, or longer periods in between drought events, compared to shorter

accumulation periods and shorter periods between drought events.

In the current manuscript, we show that the standardised groundwater time series in Figure 1 correlate well with standardised precipitation time series at different precipitation accumulation periods [R212]. These different optimal accumulation periods (the accumulation period for precipitation with highest correlation with groundwater levels at a given site) were selected for each of the groundwater time series [R212-217]. The optimal accumulation period is indeed different for the two highlighted examples (cluster C2 and C4), respectively 18 and 8 months. This difference is to account for the different autocorrelation within the groundwater time series and the natural (short/long) delay in recharge. The identified optimal accumulation periods are similar to the published results in Bloomfield & Marchant (2013), who also analysed the relation between optimal accumulation period and autocorrelation of groundwater time series. When applying these different, optimal precipitation accumulation periods, high $SPI_Q$-SGI correlations were found for the Chalk clusters (Table 1). The high correlations are not surprising, as these Chalk groundwater wells are considered near-natural [R121-126] and the high correlations confirm that differences in periods between drought events and drought occurrence in these clusters are related to driving precipitation and the hydrogeological setting [R301-311]. The slight variation in $SPI_Q$-SGI correlation within these Chalk clusters is included in the analysis when distinguishing between influenced and uninfluenced groundwater time series. The lowest $SPI_Q$-SGI correlation of each cluster (3rd column) was used as threshold to determine which groundwater time series are influenced and uninfluenced for the paired water management units [R176-178].

| Chalk clusters | Average SPIQ-SGI | Lowest SPIQ-SGI in cluster | Average optimal accumulation period (in months) |
|---|---|---|---|
| 1 | 0.78 | 0.75 | 12.6 |
| 2 | 0.82 | 0.69 | 18.2 |
| 3 | 0.81 | 0.71 | 24.0 |
| 4 | 0.73 | 0.66 | 8.0 |
| 5 | 0.75 | 0.70 | 7.5 |

**Table 1.** SPI$_Q$-SGI correlation for Chalk clusters presented in Figure 1 in manuscript of Wendt, et al. (2020)

Literature

Bloomfield, J. P. and Marchant, B. P.: Analysis of groundwater drought building on the standardised precipitation index approach, Hydrology and Earth System Sciences, 17, 4769–4787, https://doi.org/10.5194/hess-17-4769-2013, 2013.

Bloomfield, J. P., Marchant, B. P., Bricker, S. H., and Morgan, R. B.: Regional analysis of groundwater droughts using hydrograph classification, Hydrology and Earth System Sciences, 19, 4327–4344, https://doi.org/10.5194/hess-19-4327-2015, 2015.

Eltahir, E. A. B. and Yeh, P. J. F.: On the asymmetric response of aquifer water level to floods and droughts in Illinois, Water Resources Research, 35, 1199–1217, https://doi.org/10.1029/1998WR900071, 1999.

Environment Agency: Vale of St. Albans Numerical Groundwater Model Final Report, Tech. rep., 2010.

Haas, J. C. and Birk, S.: Characterizing the spatiotemporal variability of groundwater levels of alluvial aquifers in different settings using drought indices, Hydrology and Earth System Sciences, 21, 2421–2448, https://doi.org/10.5194/hess-21-2421-2017, 2017.

Lee, J. M., Park, J. H., Chung, E., Woo, N. C.: Assessment of Groundwater Drought in the Mangyeong River Basin, Korea, Sustainability, 10, 831, 2018

Ohdedar, B.: Groundwater law, abstraction, and responding to climate change: assessing recent law reforms in British Columbia and England, Water International, 42, 691–708, https://doi.org/10.1080/02508060.2017.1351059, 2017

Peters, E., Bier, G., Van Lanen, H. A. J., and Torfs, P. J. J. F.: Propagation and spatial distribution of drought in a groundwater catchment, Journal of Hydrology, 321, 257–275, https://doi.org/10.1016/j.jhydrol.2005.08.004, 2006.

Shepley, M. and Streetly, M.: The estimation of 'natural'summer outflows from the Permo-Triassic Sandstone aquifer, UK, Quarterly Journal of Engineering Geology and Hydrogeology, 40, 213–227, 2007.

Shepley, M., Pearson, A., Smith, G., and Banton, C.: The impacts of coal mining subsidence on groundwater resources management of the East Midlands Permo-Triassic Sandstone aquifer, England, Quarterly Journal of Engineering Geology and Hydrogeology, 41, 425–438, https://doi.org/10.1144/1470-9236/07-210, 2008

Tallaksen, L. M., Hisdal, H., and Lanen, H. A. V.: Space–time modelling of catchment scale drought characteristics, Journal of Hydrology, 375, 363–372, https://doi.org/10.1016/j.jhydrol.2009.06.032, 2009

Whitehead, E. and Lawrence, A.: The Chalk aquifer system of Lincolnshire, Tech. rep., Keyworth, Nottingham, contributors: J P Bloomfield, P J McConvey, J E Cunningham, M G Sumbler, D Watling, M Hutchinson, 2006.

---

## Author Comment (AC3) · 19 May 2020

This general response can be found as supplement.

Please also note the supplement to this comment:
https://www.hydrol-earth-syst-sci-discuss.net/hess-2020-22/hess-2020-22-AC3-supplement.pdf

---

## Author Response (AR1)

Dear Xing Yuan,

Thank you for the opportunity to submit a revised version of our manuscript *Asymmetric impact of groundwater use on groundwater droughts*, to be considered for publication in Hydrology and Earth System Sciences. This manuscript has Manuscript ID hess-2020-22. We are very grateful to you and the reviewers for their careful reading and insightful comments. Incorporating these changes has improved the clarity of the manuscript greatly.

We have made some significant changes to address the key concerns of both reviewers requesting additional explanation in the method and results sections. The concerns are associated with the uncertainty when using standardised indices for drought studies and in particular for groundwater studies. To address these comments, we have gathered a number of explanatory examples and rephrased sections in order to improve the descriptions and discussion of results. The major changes we have incorporated are as follows:

1. Further and detailed explanations of the used correlation analysis and identification of human influence in standardised groundwater time series including an illustrated example.
2. Addressing the variability in presented results that included a new data analysis, improved the description, and discussion of the old and new results.

In addition, we have carefully considered the suggested alternative methods provided by the reviewers and we have made every attempt to address their concerns in a revised manuscript. We provide a detailed point-by-point response to each of the reviewer's comments below. Line numbers refer to the revised manuscript.

Thank you for considering a revised version of our manuscript and we hope to hear from you soon.

Yours sincerely,

Doris Wendt (on behalf of co-authors)
* * *
Editor comments to the Author:

**Dear Authors,**

**I would like to thank for your responses to the comments. It would be useful to upload the revised manuscript, including a track changes version.**

**Looking forward to your revisions.**

**Regards**
**Xing Yuan**

Thanks for the opportunity to upload the revised manuscript. We have addressed the reviewers' comments and revised the manuscript in line with their suggestions. Please find the detailed point-by-point response below.

Reviewer(s)' Comments to Author:

**Reviewer: 1**

**This study uses a framework that consists of two approaches, and conducted a case study in UK to investigate the impact of groundwater use on groundwater droughts. Generally, the manuscript is well organized with clear logic, before I recommend it for publication, major improvements are still needed, particularly for the method they used for recognizing the presence or absence of human-influence on groundwater. Please find my specific comments below:**

Thank you for your comments. We are relieved to hear that the structure and logic of the paper was well-received. We thank you for your careful reading and have modified the manuscript along the lines of your suggestions that greatly improved the clarity of the manuscript.

**General comments:**

**R1C1: Lines 171-172: 'In this study, the presence or absence of human-influence on groundwater was determined in relation to the lowest SPIQ-SGI correlation of each near-natural reference cluster'. I think it is questionable to determine the presence or absence of human influence depending on the correlation analysis. For example, for a certain site, SGI is best correlated to SPI at short time scales. Due to human interference, the drought duration indicated SGI may become longer, leading to SGI best correlated with SPI at longer time scales. The increased time scale of SGI does not necessarily corresponds to reduced correlation of SPIQ-SGI, and the correlation may also increase. Moreover, considering the significant spatial heterogeneity of groundwater (e.g., groundwater of the monitoring sites may show different patterns from reference sites), it would be better to recognize human influences by analyzing the temporal variation of groundwater for the same site (e.g., compare the statistics of groundwater among different decades). The uncertainty derived from the method for recognizing the presence or absence of human-influence on groundwater needs to be discussed.'**

We understand the concern raised in comment 1 regarding the correlation analysis, but disagree with the assertion that the inference of abstraction effects cannot be inferred through analysis of correlations between $SPI_Q$ and SGI, and their specific assertion that correlations may increase with increased abstraction effects. We set out our arguments in more detail below and have extended section 3.2.2 as shown below (Lines 164:180) to reflect our response. We agree that uncertainty in recognising the presence or absence of abstraction effects needs to be discussed: Reviewer 2 has raised similar points (see R2C3 and R2C4). We now provide an example (in the new Supplementary Information, S2) that illustrates the method and more discussion of uncertainties at Lines L251:262 in the results section. More detailed justification for these changes is given below.

> [*L164:180*]: *Under near-natural conditions, the optimum correlation between standardised precipitation and groundwater indices ($SPI_Q$-SGI) is generally high in unconfined aquifers (Bloomfield and Marchant, 2013). Anomalies in precipitation propagate with a relatively constant delay in recharge to the groundwater, which is due to, subsurface controls on recharge, the antecedent condition of the land surface, and non-linear response of groundwater systems (Eltahir and Yeh, 1999; Peters et al., 2006; Tallaksen et al., 2009). This constant delay is included by the optimal precipitation accumulation period in the calculated $SPI_Q$-SGI correlation represents a long-term relationship for a certain site, as both the SPI and*

*SGI were calculated for a continuous 30-year period including all seasons and both anomalously dry and wet periods.*

*The $SPI_Q$-SGI correlation can be reduced when groundwater level response becomes disconnected from driving precipitation under confined conditions (Bloomfield et al., 2015; Kumar et al., 2016; Lee et al., 2018) or when groundwater abstraction changes groundwater storage and levels independent from driving precipitation (Bloomfield et al., 2015; Lorenzo-Lacruz et al., 2017; Haas and Birk, 2017). In this study, the impact of confined conditions on reducing the $SPI_Q$-SGI correlations is expected to be minimal, as only a small selection of Chalk sites are located in the semi-confined Chalk in South Lincolnshire (Table 1). On the other hand, the impact of dynamic groundwater use on $SPI_Q$-SGI correlations is expected to be significant, as long-term changes in groundwater use in the water management units resulted in a spatially heterogeneous pattern of irregular, decreasing, or increasing influence of abstraction on groundwater storage. For example, Ohdedar (2017) shows that groundwater use in the UK increased until the late 1980s and reduced afterwards with a large redistribution of where water is taken from to minimise the impacts on low flows.*

There are three main reasons why we believe that our approach is appropriate, as follows: 1) the definition and nature of SGI and SPI, and high $SPI_Q$-SGI correlations based on long-term average relationships for all groundwater levels under near-natural conditions, 2) the irregular and dynamic nature of groundwater abstraction in the water management units, and 3) consistency with the results of previous studies.

First, we would like to emphasize that correlations between standardized precipitation and groundwater time series are generally high in unconfined systems for near-natural conditions (Bloomfield and Marchant, 2013). Long-term standardised groundwater and precipitation indices (SGI and SPI respectively) were calculated for a continuous period including all seasons and both anomalously dry and wet periods. The calculated $SPI_Q$-SGI correlations represented thus a long-term average relationship between precipitation and groundwater response, not just the relationship during episodes of drought. Consequently, the suggestion of Reviewer 1 that anthropogenic influences during droughts might increase $SPI_Q$-SGI correlations at longer accumulation periods would only occur if abstraction effects were sustained for the majority of the analysis period, not just during droughts since the correlation is based on a 30-year record. We found no evidence that this has occurred in the four investigated water management units, in fact for all units if anything a decrease in overall groundwater use was found (see Table 1 in manuscript).

Secondly, there are two main reasons why the long term average $SPI_Q$-SGI correlations may be reduced. The first reason is when groundwater level response becomes disconnected from driving precipitation under confined conditions (Bloomfield et al., 2015; Kumar, et al. 2016; Lee et al., 2018). For our sites, this is not considered to be a significant issue, as only a few sites are located in sections that become increasingly confined (Southern Lincolnshire; see Table 1). The second reason for reduced $SPI_Q$-SGI correlation is the effect of groundwater abstraction. In this study, groundwater abstraction is conceptualised as exerting change in groundwater storage, and therefore groundwater levels, independent of natural changes in groundwater storage associated with driving precipitation. These changes are considered highly dynamic in both space and time, as multiple abstraction wells in a water management unit (i.e. well field) are typically used to meet water demand. We don't have quantitative information about either detailed operational practices, but there is sufficient evidence that abstraction and management practises have changed during the period of investigation.

The amount of abstracted groundwater depends on the dynamic water demand and management policies in place. For example, water demand is often seasonal with higher abstraction in spring and summer. This seasonal change in water use was previously found to reduce long-term correlations between precipitation and groundwater (Lorenzo-Lacruz et al., 2017). In addition to seasonal variation, long-term changes in groundwater abstraction show an increase up until the late 1980s nationally, since when legislation has resulted in a general reduction in abstraction, but with a redistribution of where water is taken from to minimise the impacts of surface flows (Whitehead and Lawrence, 2006; Environment Agency, 2010; Shepley et al., 2008; Shepley and Streetly, 2007; Ohdedar, 2017). Both short-term (seasonal) and long-term changes in abstractions are likely to result in a spatially heterogeneous pattern of irregular, decreasing, or increasing influence of abstraction on groundwater storage.

Thirdly, our hypothesis that this highly dynamic pattern of groundwater abstraction will result in reduced $SPI_Q$-SGI correlations matches previous research, for example by Bloomfield et al., 2015, who found lower $SPI_Q$-SGI correlations for wells that are influenced by groundwater abstractions (clusters 3 and 6). Another example of disturbance of this relationship is given by Haas and Birk (2017), who showed that correlations between precipitation, streamflow, and groundwater observations are reduced due to the interference of power plants.

Lastly, the complexity and irregularity of management practices across the study sites combined with the lack of quantitative information on abstractions have also meant that we could not do an analysis of temporal variations in groundwater response to abstraction.

Consequently, based on these considerations we feel that our working hypothesis that the varying influence of abstraction will contribute to a reduction in the long-term $SPI_Q$-SGI correlation is reasonable. We have adjusted L164:180 in the new version of the manuscript to also clarify this for the reader.

**R1C2: Section 3.2.4 Lines 185-196. The authors use the statistic variable 'Z' of the Mann Kendall test to judge whether the groundwater of the monitoring sites involves human influences. I think the statistic variable 'Z' can indicate the significance level (e.g., when |Z|>2.56, it suggests a significant trend), however, it seems arbitrary to conclude that the detected trend becomes more significant with increased value of |Z|. Fortunately the authors mentioned that both PET and precipitation present no significant trend, while groundwater level presents significant trend. This inconsistent pattern between PET /precipitation and groundwater level may imply the existence of human influence. I suggest the authors use additional methods (e.g., linear regression) to confirm the existence of human influence.**

We agree with Reviewer 1 that the description of the trend Z indicator could be improved. We have addressed this point by extending our description of the trend test methodology at Section 3.2.4 [L201:206]. We have also added additional detail to the results section 4.3 [L301:308] and modified Figure 3 to include significant and non-significant trend values. We have also changed the significance level as suggested by Reviewer 1 that resulted in slight rephrasing of the results' section (see below).

> *[L201:206]  Trends were quantified by the trend Z value showing positive or negative deviations from the null hypothesis (no trend). Positive/negative Z values indicated increasing/decreasing trend directions. |Z| values over |2:56| (α = 0.01) were considered significant. Trends in groundwater level time series were tested using a modified Mann-Kendall trend test (Mann, 1945; Kendall, 1948), which includes a modification developed by Yue and Wang (2004) to account for significant auto-correlation in the annual groundwater*

*data (Hamed, 2008). Trends in climate time series were also calculated from annual data using a standard Mann-Kendall trend test.*

*[L301:308] Significant trends in groundwater level were detected in 38% of all monitoring sites in the water management units. Of these 38%, half of the trends are upward (positive) and the other half is downward (negative) trends (Figure 3). Overall, upward trends are dominating (61% of sites including significant and non-significant trends), indicating a sustained rise in the 30-year groundwater level time series. Fewer (39% including significant and non-significant) downward trends are detected indicating sustained lowering of groundwater levels. The presence of these significant trends in groundwater is notable given the weak, non-significant, trends in the 30-year precipitation and potential evapotranspiration data (P: Z =−0.75-1.53, PET: Z=0-0.65).*

We mention in the manuscript that the significant auto- and serial correlation in the groundwater time series [L204:205] limits the application of parametric methods, such as linear regression, which is only applicable to normally distributed independent data. We tested the annual groundwater level time series and only 5 out of 170 time series are normally distributed (Shapiro-Wilk Normality Test). All others (165 time series) deviate from a normal distribution, which was also found by Bloomfield and Marchant (2013) for their groundwater time series. Therefore it seems unsuitable to apply linear regression to the majority of the groundwater dataset.

**R1C3. The time series of SGI for reference wells in Figure 1 (Section 4.1) show significant spatial heterogeneity, and their time scales vary from one site to another. For example, C2 #5 presents long time scales, while C4#9 presents short time scales. This may lead to the higher correlation between SPIQ-SGI for C2 #5 than C4#9 (see comment 1). I think the way of using correlation to judge the human influence is worth thinking**

We acknowledge the noted spatial heterogeneity by Reviewer 1. This is consistent with previously documented spatial variations in the characteristics of (autocorrelation structure and record of hydro-climatic extremes) of groundwater level time series in the Chalk by Marchant and Bloomfield (2018). However, there is no systematic evidence for higher $SPI_Q$-SGI correlations between sites with longer (C2 #5) or shorter (C4 #9) SPI accumulation periods (or autocorrelations in SGI), in fact quite the contrary, $SPI_Q$-SGI correlation has been shown to be broadly independent of SPI accumulation period. We illustrate this in the Figure below which shows $SPI_Q$-SGI correlation co-efficient as a function of SPI accumulation period (months) and SGI autocorrelation for data from this study (blue triangles and dots respectively) and for data from Bloomfield and Marchant (Table 2, 2013) (orange triangles and dots respectively) from Bloomfield and Marchant (Table 2, 2013). Consequently, we have rephrased the current explanation [L222:226] to highlight the consistency of $SPI_Q$-SGI correlations for these near-natural sites. Below this inserted text, we provide some additional evidence for this revision to the text.

*[L222:226] The optimal $SPI_Q$-SGI correlations of the near-natural wells are high on average (0.79) with a range of 0.66 to 0.89. These correlations are found using the optimal accumulation period, which accounts for delay in recharge that is different for each reference cluster. High $SPI_Q$-SGI correlations are found for both short and long accumulation periods and there was no systematic relationship between the $SPI_Q$-SGI correlation and the SPI accumulation period Q or SGI autocorrelation in the near-natural wells.*

[Figure]

**R1C4: Lines 254-264 and 315-318: The authors mentioned that 'The first pattern, apparent in Lincolnshire, Chilterns, and Shropshire, shows an increase in short drought events often found before a major drought event or during hot summers, which is probably related to an increase in water use'. However, from Fig.2 it shows that 'minor droughts C2 before major drought events' are not limited to influenced sites, similar phenomena are also observed in uninfluenced sites. Other factors such as the drought identification method, and the spatial heterogeneity of groundwater may also generate such minor droughts. It seems arbitrary to attribute such events to the increased water use and there is much uncertainty on the results.**

We agree with Reviewer 1 that the paper would benefit from an additional analysis and justification of statements related to the interpretation of the occurrence of minor droughts. Consequently, we have provided more text in the results section at Lines 268:283, including a new data analysis and figures in the Supplementary material. In Lines 284:295 we provide additional contextual information from other studies. The discussion section has, consequently, also been rephrased [344:350].

> *[268:270] Categorised influenced sites (those with SPI$_Q$-SGI correlations lower than the cluster minimum) had typically shorter drought events of a lower magnitude. The distribution of drought duration in Figure S6 shows that the majority of these additional droughts is recorded in influenced sites compared to uninfluenced sites in Lincolnshire, Chilterns, and Shropshire.*

> *[279:283] However, there was no consistency between the study areas in relation to the timing of these shorter drought events. In Lincolnshire, minor droughts occur more often during reference droughts. In the Chilterns and Shropshire, more droughts are detected prior to reference droughts (Table S8). All minor droughts are shorter than the groundwater memory (auto-correlation) suggesting that these minor droughts are less likely to be related*

*to propagated precipitation deficits, but instead are probably related to groundwater abstraction.*

*[284:295] Drought descriptions in the literature show an increase in water demand during the 1995-97, 2003-06 and 2010-12 drought (Walker and Smithers, 1998; Marsh et al., 2013; Durant, 2015). For example, Durant (2015) found that during the 1988-93 drought event evapotranspiration was exceptionally high. Impacts were mostly felt in the Chalk, particularly in regions where groundwater is the principal source of water supply where abstractions amplified the drought effects. An extreme rise in water use was reported by Walker and Smithers (1998) during the 1995-1997 drought event putting strain on drinking water supply systems in North East England. Sections of the Permo-Triassic sandstone were amongst the worst affected with drought conditions until 1998 (Durant, 2015). During the 2003-06 and 2010-12 droughts, a sudden increase in groundwater use was found that was attributed to dry weather and hot summers in the work of Marsh et al. (2007, 2013) and Durant (2015). In the work of Rey et al. (2017), low $SPI_3$ values were found in summer months for 1995, 1996, 2003-2006, and 2010-2011 highlighting exceptional dry weather that led to surface water use restrictions prior to droughts to maintain low flows. Consequently, the reduced surface water abstractions were replaced by groundwater, for which use was rarely restricted (Rey et al., 2017) resulting in lowered groundwater levels and potentially aggravating groundwater droughts.*

*[344:350] The first pattern, apparent in Lincolnshire, Chilterns, and Shropshire, shows an increase in short drought events in influenced sites that sometimes occur before a major drought event or during unusual dry period that results in a rapid increase in both surface water and groundwater use (Walker and Smithers, 1998; Marsh et al., 2013; Durant, 2015) and/or complementary groundwater use due to surface water use restrictions (Rey et al., 2017; Rio et al., 2018). We see the effect of this local increase in water use in our data in the temporarily lowered groundwater levels resulting in additional drought events. The majority of these events occur in influenced sites, but some of the (on average) uninfluenced sites also show minor droughts. Given the high correlation in these uninfluenced sites, the minor droughts seem to not disturb the long-term average correlation.*

In the new analysis described below, we show the distribution of drought duration for influenced and uninfluenced sites (see Figure 1 included in Supplementary material S6.). Table 2 in the manuscript shows that mean drought duration is lower in influenced sites and Figure 1 shows the distribution of these shorter droughts (3-5 months) for influenced sites. The spike in the distribution plot confirms the increased occurrence of minor droughts in influenced sites in Lincolnshire, Chilterns and Shropshire. These minor droughts vary slightly in duration, as expected, given the different hydrogeological settings of the water management units.

[Figure]

*Figure 1: Distribution of drought duration for classified influenced (blue) and uninfluenced (grey) sites in the four water management units. The average drought duration is highlighted by the striped vertical line in the graph (colours are matching).*

The additional data analysis also shows that not the majority, but 27 to 43 percent of the shorter droughts occur 1-24 months before the reference droughts (fourth column Table 2). In Lincolnshire, 60 percent of the shorter droughts occurs during the reference droughts compared to a smaller percentage in the Chilterns (27%) and Shropshire (23%; in fifth column Table 2). We have changed the wording in lines 278:281 to highlight the relative occurrence of the shorter drought events.

*Table 1: Duration and occurrence of minor droughts in influenced sites in Lincolnshire, Chilterns, and Shropshire. Results show that the average during is shorter than the average groundwater memory (auto-correlation).*

| Water management units | Average duration of minor droughts (in months) | Average autocorrelation (in months) | Occurrence of minor droughts 24 months before reference droughts (%) | Occurrence of minor droughts during reference droughts (%) |
|---|---|---|---|---|
| 1: Lincolnshire | 3.1 | 11.6 | 27 | 60 |
| 2: Chilterns | 3.7 | 17.3 | 34 | 27 |
| 4: Shropshire | 5.0 | 15.1 | 43 | 23 |

The minor droughts recorded in the influenced sites are also shorter than the groundwater memory (auto-correlation in third column Table 2) for all water management units suggesting that these droughts are not related to a natural deficit in groundwater due to a propagated precipitation

deficit, but to abstraction influence. Contextual information shows that increased water use has amplified existing droughts and that increased groundwater use was related to periods of exceptional dry weather. We have rephrased the text in order to clarify the sources of this contextual information [L279:295].

**R1C5: Lines 322-323, The authors mentioned 'We see the effect of this local increase in water use in our data in the temporarily lowered groundwater levels, resulting in additional drought events'. Could you provide additional information on the evolution process of water use and droughts, e.g., show the time series of both water use data and groundwater levels in one figure.**

We agree with Reviewer 1 that it would be interesting to analyse both water use data and groundwater level variations in one figure. Unfortunately, this is not possible given the unavailability of abstraction records [L59:60]. The unavailability of detailed, time-varying records oft abstraction is the primary reason for developing the methods here to infer abstraction influence [L62:63]. We have, however, provided an example of four groundwater hydrographs in the Chalk for which the first is categorised as near-natural, and the other three are groundwater monitoring sites (see S2 in R2C3). Out of these three monitoring sites, the first site shows a high correlation with the accumulated SPI, hence classified as uninfluenced. The other two sites have a low correlation with accumulated SPI and have an irregular, spikey hydrograph that also shows the temporarily lowered SGI values despite normal or above-normal SPI. These two wells are assumed to be continuously influenced by abstraction resulting in a lower $SPI_Q$-SGI correlation.

**R1C6: Lines 115-118, The authors failed to illustrate how they calculate SGI clearly. For example, which probability distribution was employed to fit the groundwater series. Whether the impact of data seasonality was considered when calculating SGI? More details on the computation of SGI should be added.**

We have changed the phrasing in lines 133-136 explaining the assigned SGI value and calculation of SGI. Note that SGI is calculated here using the non-parametric method of Bloomfield and Marchant (2013) so no assumptions about distributions were made.

> *[L133:136] Monthly groundwater observations were grouped for each calendar month and within each month observations were ranked and assigned a SGI value based on an inverse normal cumulative distribution of the data. No distribution was fitted, but SGI values were assigned to monthly observations accounting for seasonal variation within the calendar year.*

**R1C7: Lines 120-121: '208 sites have been included in the analysis, 39 are reference sites and 170 monitoring sites. ' Here '208 sites' should be '209 sites (170+39=209)'.**

We thank Reviewer 1 for spotting this mistake. It should be 209. In total, there are 39 near-natural reference wells areas (9 in the PT sandstone and 30 in the Chalk). There are 170 groundwater monitoring sites divided over the four water management units (see Table 1 first column).

> *[L118:119] 209 sites have been included in the analysis, 39 are reference sites and 170 monitoring sites.*

**R1C8: Lines 130-131: How do you fill the missing sequences, using the time series of adjacent sites? Details on the linear interpolation method should be supplemented. Besides, sites with missing data more than 6 months would be removed directly?**

We interpolated the missing data from the last measured groundwater observation to the next linearly if that missing sequence was not longer than 6 months, as previously applied in the work of

Tallaksen and Van Lanen, (2004) and Thomas et al., (2016). Groundwater sites with missing sequences longer than 6 months were indeed removed from the dataset prior to the analysis. The text has been revised as such at L127-129.

> *[L127-129] Missing data were linearly interpolated from the last observation to the next observation in case of short sequences of missing data (less than 6 months) (Tallaksen and Van Lanen, 2004; Thomas et al., 2016).*

**R1C9: The current form of Fig. 2 makes it difficult to judge the impact of human influences. The authors could add the time series of SGI for the monitoring sites so that readers can easily find human influenced periods**

We thank Reviewer 1 for this final comment, but in light of the newly added examples in S2 showing four SGI time series to illustrate the method (R2C3), we don't think is necessary to add more SGI time series to Figure 2. This is because, the design of Figure 2 is so that the timing and magnitude of groundwater droughts can be overviewed at glance. We wanted to highlight that groundwater droughts observed in sites with a reduced $SPI_Q$-SGI correlation differ in timing and magnitude. To transform this graph into time series would require a figure capturing 170 time series that would, in our opinion, result into more confusion than the highlighted droughts occurring in these 170 time series.

In addition to this, we would like to highlight that there are no specific 'human-influenced periods' identified in the investigation period. We have contextual information about the overall water use that changes in time showing that the aquifers have continuously regulated from the 1960s until now (Ohdedar, 2017).
* * *
**Reviewer: 2**

**This paper investigates the impact of groundwater use on groundwater drought for a case study in the UK.**

**Overall, I found the paper to be well-written, with some interesting results supported by some nice figures. The work represents a useful contribution to better understand how groundwater use affects groundwater drought and how groundwater levels have changed over time in the UK.**

**My main suggestions for the paper are to improve the clarity of the methods and reflection of the variability in groundwater levels. I agree with reviewer 1 that there is a lot of uncertainty in the results and some of the links between groundwater use and groundwater drought are somewhat arbitrary. I sympathise with the authors as I know how difficult it is to gain groundwater use data that may help make these findings more robust, nevertheless, I believe the authors could make more efforts to discuss the limitations of their results and report the uncertainty/variability in their results**

We thank Reviewer 2 for their careful reading and constructive comments. We have pleased to hear that the contribution of the study is useful and have addressed uncertainty and limits of the current methods. Please find our point-by-point responses to comments below.

**R2C1: In the methods/discussion please add some comment on the choice of gamma distribution used to calculate SPI. Other studies have shown that this is often not the**

**most appropriate distribution for precipitation data and it would be good to discuss the impacts of this (see Svensson et al. 2017 for example).**

We thank Reviewer 2 for their comment. We now tested the alternative distributions for the SPI calculation and have included additional phrases to comment on the choice of gamma distribution used to calculate the SPI. The uncertainty has now been addressed more explicitly by improving the phrasing in L110:116.

However, we would like to emphasise that the SPI is primarily used in combination with the SGI to find the optimal correlation between the SPI and SGI. For this correlation, primarily long accumulation periods (> 12months) of the SPI were used (see the mean of optimum accumulation periods in L226:230 for near-natural wells and the range of accumulation periods in S3 in the supplementary material). Considering the use of long accumulation periods, the `best' fitting distribution varies (Svensson et al. 2017). High rejection rates are found for multiple distributions (Stagge et al. 2015), which suggests we need to test which distribution performs best. We have tested different distributions for a subset of the data, shown below the inserted text.

> *[L110-116] Precipitation estimates were converted into standardised precipitation indices (SPI) following the method of McKee et al. (1993). A gamma distribution was fitted to precipitation estimates and alternative distributions were also tested (Normal, Pearson III, and Logistic). Considering the use of SPI to account for delayed recharge, a large range of accumulation periods of precipitation (1 to 100 months) was calculated in order to find the optimal correlations between precipitation and groundwater time series. For this particular use of the SPI, the 'best' fitting distribution varies (Svensson et al., 2017). Alternative distributions showed minimal differences in the computed correlations between standardised precipitation and groundwater time series, hence we decided therefore to use the gamma distribution.*

The additional test is performed on a subset of the total dataset (45 precipitation grids matching to groundwater monitoring sites in the Chilterns). Three alternative distributions were tested: Normal, Pearson III, and Logistic distribution and results are presented in a similar way to Figure 5d in Svensson et al., (2017) (Figure 2). Figure 2 shows an example of a $SPI_{15}$ in which a slight variation is seen in the calculated SPI values during droughts using different distributions. This variation did, however, not result in higher or lower $SPI_Q$-SGI correlation. For the subset of the data (45 monitoring sites), the range of correlations using the Gamma distribution was 0.41-0.89 with a mean of 0.794. The mean of the calculated correlation remained the same when using different distributions (Normal, Pearson III, and Logistic distribution). The range of the 45 correlations showed minimal changes compared to Gamma distribution (0.41-0.89): 0.40-0.89 (Normal & Logistic), 0.40-0.90 (Pearson III). Reviewing the minimal change in $SPI_Q$-SGI correlation, we think that the use of alternative distributions instead of the current distribution (gamma) would not change the results of this study given the use of the SPI only in the correlation analysis.

[Figure]

*Figure 2: SPI₁₅ computed using different distributions for a precipitation estimate located in the Chilterns (the location corresponding to groundwater site (SP90.27).*

**R2C2**: **From the methods section, it seems that you compare the SPI from a single grid cell with the corresponding groundwater well location (this should be clarified in the text). It would be good to add to the discussion the impacts of comparing a 1km2 grid cell of SPI with SGI that is a product of a regional groundwater aquifer system and regional rainfall patterns.**

We agree with Reviewer 2 and rephrased the text to clarify our approach (L104:109) and provide additional context below.

> *[L104:109] We aggregated daily potential evapotranspiration estimates to monthly sums. For both gridded datasets (GEAR and CHESS) grid cells were extracted corresponding to groundwater well locations. The 1km² gridded precipitation and potential evapotranspiration sums were compared to the monthly groundwater observations of the same location. This point-scale comparison assumes that the influence of precipitation is largest surrounding the groundwater monitoring site (Bloomfield and Marchant, 2013; Bloomfield et al., 2015; Li and Rodell, 2015; Kumar et al., 2016).*

The regional extent of groundwater recharge varies and the precise extent of this recharge area associated with a given observation borehole is often unknown. In contrast to surface water boundaries, there is no consistent source of information regarding the recharge area for groundwater monitoring sites in the UK in either the Hydrometric Register (Marsh and Hannaford, 2008) or the water management units. The unknown recharge area is a common uncertainty for groundwater studies and other studies have either used a regional aggregate to overcome this unknown recharge area (Haas et al., 2018) or used a point-scale analysis under the assumption that the influence of precipitation is largest surrounding the groundwater monitoring site (Bloomfield and Marchant, 2013; Bloomfield et al., 2015; Li and Rodell, 2015; Kumar et al., 2016). Even though a regional precipitation product would potentially result in a more accurate correlation, it could also

introduce larger uncertainties given the unknown extent. In addition to this, high correlations between $SPI_Q$-SGI have been previously obtained using the point-based precipitation estimates in different climate regions and by different authors (Bloomfield and Marchant, 2013; Bloomfield et al., 2015; Li and Rodell, 2015; Kumar et al., 2016). Also in our study, high correlations are found for near-natural wells and the majority of groundwater monitoring wells [L222:226 and L231:235]. Therefore, considering the unknown recharge area and reasonable results, we don't propose to modify our methodology, which is consistent with previous studies using point-based precipitation estimates.

**R2C3: The methods (in places) were not clear – in particular, the SPI_SGI correlations and the use of the near-natural wells, uninfluenced and influenced monitoring sites. It would be useful to have a worked example of how the SPI-SGI correlations work in practice (showing an example for two sites – one influenced and one non-influenced and how they compare to the near natural reference cluster). It would also be useful to have a map of the influenced and non-influenced wells (this is maybe already included in Figure 1 but this figure is quite busy so it is hard to tell!)**

In response to R2C3, we have added an illustration of the $SPI_Q$-SGI correlation methodology to the Supplementary information (S2) using four wells to show the $SPI_Q$-SGI correlations in a single water management unit for a near-natural reference site and three groundwater monitoring sites (influenced and uninfluenced) and referred to this illustrated example in the section 3.2.2.

> [L185:186] An illustrated example is provided in Figure S2 showing SGI time series of a near-natural reference site and three groundwater monitoring sites.

We agree with Reviewer 2 that it would be very interesting to show spatial patterns of detected influenced wells. We had considered mapping locations of influenced wells, but analysing and explaining spatial patterns in such maps would require detailed knowledge of the hydrogeological setting of each monitoring well and records of abstraction wells close by – information that we don't have. There is no consistent spatial pattern based on annual maximum abstraction licences. We expect that some wells are used episodically or not at all, while others are used regularly resulting in a highly variable picture. Given the unexplained spatial patterns, complex local hydrogeological structure, and the unknown use of the abstraction wells, we have not included these manuscript, as more information is required to explain the spatial patterns.

**R2C4: Reflection of uncertainty/variation**
**There is a lot of variation in the groundwater levels between sites and this needs to be better reflected in the results. I suggest that the authors report the min/max or 5th/95th percentile of their results alongside the average in Table 2 and elsewhere in the text.**

As suggested by Reviewer 2, we have amended Table 2 to include the 5th and 95th percentiles of the duration, magnitude and frequency of groundwater droughts at the uninfluenced and influenced sites for each water resource management unit.

We have also included primary reasons for the variation in the groundwater levels [L241-245], as the groundwater level observations are set in a range of different hydrogeological settings and drought events vary in timing, intensity and duration, as groundwater droughts are episodic. On top of the spatial and temporal differences, human-influence on groundwater level variations change in time, which results in the variation in Table 2. We improved the phrasing in the results section describing these three different facets in the drought characteristics [L251-253].

> [L241-245] Groundwater droughts observed in the reference clusters reflect both spatial and temporal variation due to driving precipitation and hydrogeological setting. In general, the

> *four UK-wide droughts (1988-1993, 1995-1998, 2003-2006, and 2010-2012) are reflected in near-natural groundwater time series. Spatial patterns in driving precipitation, however, result in variable groundwater drought occurrence (Figure 1).*

> *[L251-253] On a smaller scale in the water management units, average drought characteristics (duration in months, magnitude in accumulated SGI over the drought period, and frequency) for monitoring sites show differences due to abstraction influence, which we have classified in, on average, uninfluenced and influenced sites, see Table 2.*

In addition, to better illustrate the variability in the drought characteristics between uninfluenced and influenced sites we have introduced a new set of distribution figures in the Supplementary Information (S5-S7) and provided additional explanatory text at Lines 255:262.These distribution figures show the difference and overlap between influenced and uninfluenced sites.

> *[L255-262] Droughts are observed twice as often in influenced compared to uninfluenced sites in Lincolnshire and Chilterns, but this difference is smaller in Shropshire. The distribution of recorded drought frequency (Figure S5) shows that the difference between on average influenced and uninfluenced sites is less pronounced in Lincolnshire and Shropshire. Table 2 shows that the average drought duration of influenced sites exceeds the duration in uninfluenced sites in the Midlands. Longer and more intense groundwater droughts occurred less often in influenced sites, which is in contrast with the other water management units. The distribution of recorded drought frequency (Figure S5) shows a majority of sites recording fewer droughts and some sites that record a higher frequency. On average, this results in a small difference between the influenced and uninfluenced sites.*

**R2C5: Like Reviewer 1, I am somewhat sceptical of attributing the shorter droughts in Lincolnshire, Shropshire and the Chilterns to water use and/or hotter Summers. Firstly the years that were identified in L263-265 did not have particularly hot summers (or this is certainly not consistent for these years) and many of these drought events can also be identified in the uninfluenced wells. These uncertainties need to be reflected in the discussion or the methods for identification need to be more robust.**

We understand the concern raised regarding the attribution of shorter droughts to increased water use by Reviewer 2 and earlier by Reviewer 1 (R1C4). We have rephrased the result and discussion section [267:295, and 344:350, see R1C4] and have included the new data analysis in the supplementary material (S6 and Figure 1 in this rebuttal). Reviewer 2 is right to note that shorter (or minor) droughts are also observed in uninfluenced sites. However, the distribution graphs of recorded drought frequency and duration in groundwater monitoring sites show that the majority is in uninfluenced sites. We have also provided additional contextual information regarding the reported increased water use [284:295, see R1C4].

**Minor Comments**
**R2MC1. The abstract is quite long – I would shorten it and just highlight the key results. Currently, your more interesting results get a little lost in all the text.**

We agree with Reviewer 2 and we have shortened the Abstract into the following:

> *[L1:19] Groundwater use affects groundwater storage continuously, as the removal of water changes both short-term and long-term groundwater level variation. This has implications for groundwater droughts, i.e. a below-normal groundwater level. The impact of groundwater use on*

*groundwater droughts, however, remains unknown. Hence, the aim of this study is to investigate the impact of groundwater use on groundwater droughts in the absence of actual abstraction data adopting a methodological framework that consists of two approaches. The first approach compared groundwater droughts at monitoring sites that are potentially influenced by abstraction to groundwater droughts at sites that are known to be near-natural. Observed groundwater droughts were compared in terms of drought occurrence, magnitude, and duration. The second approach investigated long-term trends in groundwater levels in all monitoring wells. This framework was applied to a case study of the UK using four regional water management units, in which groundwater is monitored and abstractions are licensed. Results show two, asymmetric, responses in groundwater drought characteristics due to groundwater use. The first response is an increase of shorter drought events, and is found in three water management units where long-term annual average groundwater abstractions are smaller than recharge. The second response, seen in one water management unit where groundwater abstractions temporarily exceeded recharge, is a lengthening and intensification of groundwater droughts. Analysis of long-term (1984-2014) trends in groundwater levels shows mixed, but generally positive trends, while trends in precipitation and potential evapotranspiration are not significant. The generally rising groundwater levels are consistent with changes in water use regulations and with an overall reduction in abstractions during the period of investigation. We summarised our results in a conceptual typology that illustrates the asymmetric impact of groundwater use on groundwater drought occurrence, duration, and magnitude. The long-term balance between groundwater abstraction and recharge plays an important role in this asymmetric impact, which highlights the relation between long-term and short-term sustainable groundwater use.*

**R2MC2. P3 L80 It would be good to name these four water management units in the text**

Agreed. We have included the names of the water management units in now L74:75.

> *[L74:75] The UK case study consists of four water management units (1: Lincolnshire, 2: Chilterns, 3: Midlands, 4: Shropshire) across the Chalk and Permo-Triassic sandstone aquifers that are the two main aquifers in the UK (Figure 1).*

**R2MC3. P5 L117 – What accumulation periods did you calculate SPI over, you need to be more specific here.**

Agreed. We have included the full range of accumulation periods in the new version of the manuscript.

> *[L112:114] Considering the use of SPI to account for delayed recharge, a large range of accumulation periods (1 to 100 months) in order to find the optimal correlations between precipitation and groundwater time series.*

**R2MC4. Table 1 – what time period were the long term precipitation and PET calculated over? It would be good if this was consistent with the time periods used in your study.**

We thank Reviewer 2 for pointing this out, as the long-term precipitation and PET was taken from the Mansour and Hughes (2018) study and based on daily data from 1962 to 2016. We have now clarified that in the table caption.

**R2MC5. P7 L194 – Were these the climate time series from a single grid cell?**

This data is indeed from the same climate datasets using the same extracted grid cells. We have now clarified this by rephrasing L197:200.

> [L197:200] Hence, an additional trend test was introduced to compare trends in annual (averaged for each calendar year) groundwater levels to climate data (precipitation and evapotranspiration) that were extracted for grid cells corresponding to groundwater well locations from the GEAR and CHESS datasets (Tanguy et al., 2016; Robinson et al., 2016)

**R2MC6. Table 2 - I was a little surprised that the average drought frequency for the Midlands cluster is significant when the values are quite similar (9.5 for uninfluenced and 9 for influenced) – is this correct?**

We agree with Reviewer 2 that this is an interesting result and checked this before submission. The averaged difference in drought frequency is indeed statistically significant, which is now clearer when looking at the distribution the spread of the data in S5.

**R2MC7. Section 4.3 – in this section you don't distinguish between 'influenced' and 'uninfluenced' wells. It would be useful know whether the strong trends are just in the 'influenced' wells? If they are not, then your 'uninfluenced' wells may be more affected than suggested.**

We thank reviewer 2 for noting this difference between sections and we acknowledge that this topic did not receive much attention in the manuscript. We have explicitly stated this now in the abstract, methods and results [L7:8, 194, 301].

We have indeed not distinguished between influenced and uninfluenced sites and this is because the methods to categorise influence of abstraction are not designed for the identification of trends or long-term changes in groundwater levels. We have added this to the method section now [L196:198].

> [L196:198] This trend test contributes to the first approach, as the SGI and $SPI_Q$-SGI correlation analysis do not specifically account for trends in groundwater time series that could result in significant trends going unnoticed.

The trend results correspond with the categorisation of influenced and influenced sites. Most uninfluenced sites (75%) have a non-significant trends compared to most influenced sites (72%) that have a significant trend. From the uninfluenced sites, only a small percentage (5%) has a negative trend. These sites indicate an indirect influence of abstraction nearby the groundwater monitoring and time series show both a downward trend and episodic drought events that align with an accumulated SPI signal. Interestingly, 20% of the uninfluenced sites have a significant positive trend. Investigating the drivers of these significant positive trends in groundwater levels would be interesting, although beyond the scope of the current study.

**Additional references in rebuttal:**

[revised manuscript text omitted]